# Learning Large-scale Neural Fields via Context Pruned Meta-Learning

**Jihoon Tack**[1], **Subin Kim**[1], **Sihyun Yu**[1], **Jaeho Lee**[2], **Jinwoo Shin**[1], **Jonathan Schwarz**[3]

[1]Korea Advanced Institute of Science and Technology
[2]Pohang University of Science and Technology
[3]University College London
{jihoontack,subin-kim,sihyun.yu,jinwoos}@kaist.ac.kr
jaeho.lee@postech.ac.kr, schwarzjn@gmail.com

## Abstract

We introduce an efficient optimization-based meta-learning technique for large-scale neural field training by realizing significant memory savings through automated online context point selection. This is achieved by focusing each learning step on the subset of data with the highest expected immediate improvement in model quality, resulting in the almost instantaneous modeling of global structure and subsequent refinement of high-frequency details. We further improve the quality of our meta-learned initialization by introducing a bootstrap correction resulting in the minimization of any error introduced by reduced context sets while simultaneously mitigating the well-known myopia of optimization-based meta-learning. Finally, we show how gradient re-scaling at meta-test time allows the learning of extremely high-quality neural fields in significantly shortened optimization procedures. Our framework is model-agnostic, intuitive, straightforward to implement, and shows significant reconstruction improvements for a wide range of signals. We provide an extensive empirical evaluation on nine datasets across multiple multiple modalities, demonstrating state-of-the-art results while providing additional insight through careful analysis of the algorithmic components constituting our method. Code is available at https://github.com/jihoontack/GradNCP

## 1 Introduction

Neural fields (NFs), also known as implicit neural representations (INRs), have emerged as a new paradigm for representing complex signals as continuous functions (e.g., the implicit pixel coordinate to feature mapping $(x, y) \mapsto (r, g, b)$ of a 2D-image) parameterized by neural networks [43, 60]. In contrast to multi-dimensional array representations, this approach has the benefits of allowing querying at arbitrary resolutions or from arbitrary viewpoints [39, 52], removing modality-agnostic architectural choices for downstream tasks (e.g., compression [10, 48, 49], classification [11, 2] or generative modeling [55, 68]), representing a more compact parameterization and providing an elegant framework for dealing with data unsuitable for discretization (e.g., vector fields). Thus, the approach has shown great promise in modeling diverse forms of data, including images [38], videos [5], 3D scenes [43], voxels [11], climate data [25], and audio [12].

Unfortunately, fitting even a single NF to a signal is often prohibitively expensive (e.g., more than a half GPU day for a single HD video [30]), limiting the field's ability to learn NFs at scale for a large set of signals. To alleviate this issue, optimization-based meta-learning has recently emerged [53, 61], demonstrating rapid improvements in learning efficiency and thus quickly becoming a standard technique in NF research. Meta-learning allows signals to be represented at good quality within only a handful of optimization steps by replacing *tabula rasa* optimization from a hand-picked

37th Conference on Neural Information Processing Systems (NeurIPS 2023).

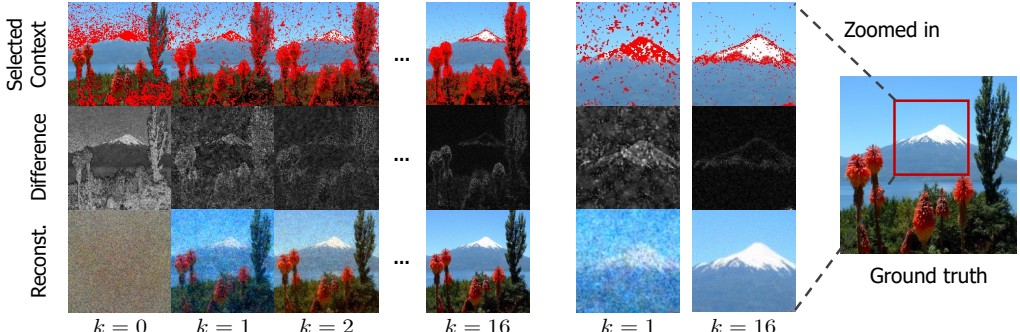

Figure 1: Online context point selection learns interpretable curricula: Visualization of selected context points (top), residuals relative to the original signal (middle), and the reconstruction (bottom) of GradNCP trained on ImageNet. Selected coordinates are highlighted in red (top) where our selection scheme focuses on only 25% of available points at each of the $k$ adaptation steps. GradNCP first focuses on modeling global structure while subsequently learning high-frequency details.

with an optimized initialization closer to minima reachable within a few steps of gradient descent. This is achieved by defining a second-order optimization problem over the initialization and other meta-parameters, computed through the iterative learning process of individual tasks, all started from a shared initialization [1, 15]. In the context of this paper, the *inner learning task* is the fitting of an NF to a specific signal (e.g., a single audio clip), while *the outer problem* corresponds to learning an initialization suitable for a whole set of signals of the same type (e.g., an entire speech dataset).

While promising in principle, existing optimization-based meta-learning schemes suffer from problems of their own, primarily poor scaling with the dataset size (often called *context set*) used in an inner learning task. This issue is particularly severe for NFs, with a single $1024 \times 1024$ image interpreted as a context set of well over a million function evaluations (i.e., $((x, y), (r, g, b))$ pairs), forming a dataset comparable in scale to ImageNet [8]. This is problematic since firstly, existing techniques are designed for the entire dataset be used in each inner optimization step, and secondly, the number of necessary inner steps required for adequate performance is known to increase with the context size. As both the computational graph and intermediate computations (e.g., gradients and activations) are stored in memory for back-propagation w.r.t. to the parameters of the outer problem, meta-learning at scale quickly becomes prohibitively expensive, particularly in terms of memory requirements. Moreover, default optimization-based meta-learning is known to suffer from myopia [16, 66], i.e., the tendency for plasticity to drastically decrease when the inner learning problem is run for more than the number of steps the initialization was optimized for. This strongly decreases the appeal of meta-learning when extremely high-quality NF representations are desired.

In this paper, we investigate whether this high computational burden can be alleviated by changing the form of the inner learning problem. In particular, we draw inspiration from curriculum learning [21] and recent interest in data pruning [44, 58], selecting a relevant subset of data points non-uniformly at each inner learning iteration to reduce the memory, and more crucially, reducing myopia through long horizon optimization made by the saved memory.

**Contributions.** We present a novel optimization-based meta-learning framework, coined *Gradient Norm-based Context Pruning (GradNCP)*, designed for large-scale problems and thus particularly suitable for learning complex NFs. Specifically, GradNCP involves online data pruning of the context set based on a new score, making memory efficiency the primary objective of our algorithmic design. By focusing learning on the most important samples, GradNCP effectively implements an automated curriculum, rapidly extracting the global structure of the signal before dedicating additional iterations to the representation of high-frequency details, resembling a hand-crafted technique for efficient NF training in previous literature [33] (see Figure 1 for an illustration of this phenomenon). As this allows the computational graph to be released for all unused context points, the resulting memory savings enable longer adaptation horizons leading to improved meta-initializations and crucially, make it possible to train on higher-dimensional signals.

To realize efficient context pruning, we argue for two key considerations: (a) the minimization of information loss caused by the context pruning by careful selection based on per-example scoring statistics (b) the correction of any remaining performance degradation and avoidance of the afore-

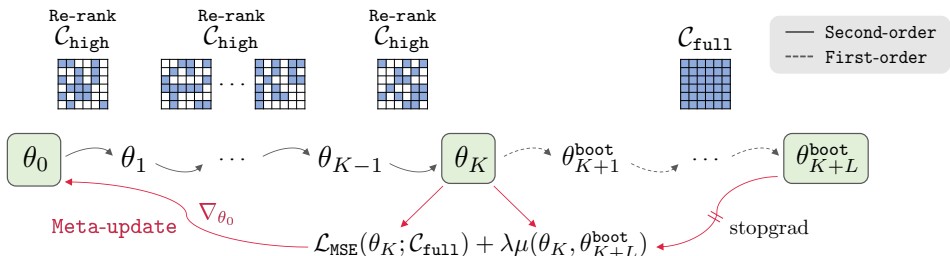

Figure 2: Computational diagram of GradNCP. A meta-learned initialization $\theta_0$ is adapted for $K$ steps to obtain $\theta_K$, re-ranking and pruning the context set $\mathcal{C}_{\text{high}}$ at each step for memory efficiency. Subsequently, we obtain a bootstrapped target $\theta_{K+L}^{\text{boot}}$ through $L$ additional steps of (reduced-memory) optimization using the *full context set* $\mathcal{C}_{\text{full}}$. Meta-optimization then minimizes both reconstruction error and the distance between the two parameter states. $\theta_0$ is thus updated to allow minimization of this distance in the original $K$ steps, correcting for context pruning and reducing myopia.

mentioned myopia phenomenon, allowing for longer optimization horizons using the full context at test time (when memory requirements are lower). To this end, we propose the following techniques:

1. *Online automated context pruning*: As large datasets involve a range of information content in its examples (e.g., in long-tail distributions), we hypothesize that learning can be greatly accelerated by focusing each training step on the examples with highest expected model improvement. To achieve this, we approximate the overall effect of updating the model on each example by the norm of the last layer gradient, an extremely cheap operation that results in virtually indistinguishable performance when compared to significantly more expensive methods while outperforming random sub-sampling by a large margin. Furthermore, we show how this technique is particularly suitable for meta-learning, demonstrating how the learning dynamics increase the correlation between the full gradient and our last layer approximation when compared to the random initialization.

2. *Bootstrap correction*: To counteract possible information loss introduced by the context pruning, we propose to nudge parameters obtained through optimization with a pruned context set to be as close as possible to an informative target obtained using the full set. We achieve this by constructing the target through continued adaptation using the *full context* and then defining a meta-loss involving the minimization of a distance between the two parameters. This acts both as a correction term while also reducing myopia, enabling advanced test-time adaptation. Importantly, this correction introduces significantly less computational overhead than simply extending meta-learning, as the target is deliberately generated without the need to save any intermediate gradients (i.e., meta-optimization is oblivious to the bootstrap generating procedure).

3. *Test-time gradient scaling*: Finally, we introduce a gradient scaling technique that allows the use of full context sets at meta-test time (when memory consumption is a significantly smaller concern) enabling the learning of the highest quality NFs while continuing to benefit from significant speed-ups afforded by meta-learning.

We verify the efficacy of GradNCP through extensive evaluations on multiple data modalities, including image, video, audio, and manifold datasets. We exhibit that our GradNCPconsistently and significantly outperforms previous meta-learning methods. For instance, measured by peak signal-to-noise ratio (PSNR), GradNCP improves prior state-of-the-art results by $38.28 \rightarrow 40.60$ on CelebA [36], and $28.86 \rightarrow 35.28$ on UCF-101 [57]. GradNCP can also lead to accelerated training, reaching the same performance as existing meta-learning methods whiled reducing wall-clock time by $\approx 45\%$. Moreover, we show that GradNCP can meta-learn on extremely high-dimensional signals, e.g., $256{\times}256{\times}32$ videos, where prior works suffer from a memory shortage on the same machine.

## 2 GradNCP: Meta-Learning via Gradient Norm-based Context Pruning

### 2.1 Meta-Learning Neural Fields

We consider the problem of learning neural fields (NFs) for $N$ signals $\mathbf{s}_1, \ldots, \mathbf{s}_N$ by taking a functional view of each signal $\mathbf{s}_i$, represented by a context set $\mathcal{C}^{(i)} := \{(\mathbf{x}_j, \mathbf{y}_j)\}_{j=1}^M$ consisting of $M$ coordinate-value pairs $(\mathbf{x}_j, \mathbf{y}_j)$ with $\mathbf{x}_j \in \mathbb{R}^C$ and $\mathbf{y}_j \in \mathbb{R}^D$. For each signal $\mathbf{s}_i$, we aim to

find parameters $\theta^{(i)}$ for the NF $f_{\theta^{(i)}} : \mathbb{R}^C \to \mathbb{R}^D$ which reconstructs $\mathbf{s}_i$ when evaluated on the full coordinate set (we also consider the generalization setup in Section 3.3). To learn NF, we minimize the mean-squared error (MSE) loss, i.e., $\ell(\theta; \mathcal{C}^{(\cdot)}) = \mathcal{L}_{\texttt{MSE}}(\theta; \mathcal{C}^{(\cdot)}) := \frac{1}{M} \sum_{j=1}^{M} ||f_\theta(\mathbf{x}_j) - \mathbf{y}_j||_2^2$. However, this requires a large number of optimization steps (on the order of 10ks for even moderately complex signals) thus limiting the development of NFs for large databases.

A popular approach for this acceleration [61] relies on model-agnostic meta-learning (MAML, [15]) by defining an optimization process over the initialization $\theta_0$ with a given step size $\alpha > 0$:

$$\theta_0 = \arg\min_{\theta_0} \frac{1}{N} \sum_{i=1}^{N} \ell(\theta_0 - \alpha \nabla_{\theta_0} \ell(\theta_0; \mathcal{C}^{(i)}); \mathcal{C}^{(i)}), \tag{1}$$

where the inner optimization is iterated for $K$ steps (typically $K << 100$) before a meta-optimization step w.r.t $\theta_0$ is taken. While mini-batch SGD is commonly used for the optimization w.r.t $\theta_0$, the entire context set $|\mathcal{C}^{(i)}| = M$ (for NFs in the order of 100ks) is required for the update in each of the inner $K$ steps (a legacy of MAML's predominant use in few-shot learning) with both intermediate computations and the computational graph maintained in memory, causing significant memory concerns. Unfortunately, both simple sub-sampling of $\mathcal{C}^{(i)}$ at each step or common first-order simplifications of the MAML objective [15, 41] have been shown to significantly under-perform in the NF context [11, 12], thus leaving practitioners with no easy solution, forcing a sub-optimal reduction of $K$ to single digits or abandonment of the meta-learning paradigm altogether.

## 2.2 Gradient Norm-based Online Context Pruning

To alleviate this memory burden, we propose an *online* context pruning strategy, resulting in the choice of a context subset $\mathcal{C}_{\texttt{high}}^k \subset \mathcal{C}_{\texttt{full}}$ at each step $k$ according to some scoring function $R_k(\mathbf{x}, \mathbf{y})$ evaluated on each example $(\mathbf{x}, \mathbf{y}) \in \mathcal{C}_{\texttt{full}}$. We denote this pruned context set $\mathcal{C}_{\texttt{high}}^k$ to distinguish from the full context $\mathcal{C}_{\texttt{full}}$ and define $\mathcal{C}_{\texttt{high}}^k := \text{TopK}(\mathcal{C}_{\texttt{full}}; R_k, \gamma)$ where TopK is a function returning the $\gamma |\mathcal{C}_{\texttt{full}}|$ elements with highest $R_k$ score. Here $\gamma \in (0, 1)$ is a hyper-parameter controlling the trade-off between (expected) performance and memory demands. We postulate that $R_k(\mathbf{x}, \mathbf{y})$ rank examples according to their effect on the overall loss when being used to update the model in the inner loop of Eq. (1), hypothesizing that a suitable $R_k$ ought to improve on random sub-sampling.

**Estimating sample importance via one-step adaptation.** The most direct but highly impractical method full-filing the requirement posed on $R_k$ is the curriculum-learning metric target-prediction gain (TPG) [21], defined as $R_k^{\text{TPG}}(\mathbf{x}, \mathbf{y}) = \ell(\theta_k, \mathcal{C} \setminus (\mathbf{x}, \mathbf{y})) - \ell(\theta_k', \mathcal{C} \setminus (\mathbf{x}, \mathbf{y}))$, i.e. the direct measurement of the reduction in loss on all other training examples after updating the full network $\theta_k$ to $\theta_k'$ using $(\mathbf{x}, \mathbf{y})$, thus requiring $\mathcal{O}(2M + M(M-1)) = \mathcal{O}(M^2)$ operations (A forward and backward pass per example and evaluation on all other $M - 1$ examples) rendering this metric inconsistent with the primary objectives of this work. A simpler alternative is the related self-prediction gain (SPG) $R_k^{\text{SPG}}(\mathbf{x}, \mathbf{y}) = \ell(\theta_k, \{\mathbf{x}, \mathbf{y}\}) - \ell(\theta_k', \{\mathbf{x}, \mathbf{y}\})$ approximating $R_k^{\text{TPG}}$ by measuring learning progress only on the example used to update the model, reducing the operations to linear complexity $\mathcal{O}(3M)$ (forward pass, model update, another forward pass). While undoubtedly a simplification, this metric has increased appeal for reconstructing NFs where we do not require generalization to previously unseen data in a fashion. Nevertheless, despite significant complexity reduction relative to $R_k^{\text{TPG}}$, the $3M$ operations of $R_k^{\text{SPG}}$ remain impractical.

Motivated by recent studies that emphasize the critical role of the last output layer in meta-learning NFs [70, 48, 49] and recognizing the relative inexpensiveness of the operation, we instead focus on an update $\theta_k'$ from $\theta_k$ involving *only the last layer*, leading to $\mathcal{O}(M)$ complexity. We then further enhance efficiency by using a first-order Taylor approximation, showing SPG under this update to be well approximated by the squared gradient norm of the last layer. Specifically, letting $\theta_k^{\text{base}}$ be the set of parameters excluding the final layer and $g_k = [\mathbf{0}, \nabla_{W_k} \ell, \nabla_{b_k} \ell]$ a padded gradient involving zeros for all but the last layer's weights $W_k$ and biases $b_k$, we start from the SPG metric:

$$\ell\big(\theta_k; \{(\mathbf{x}, \mathbf{y})\}\big) - \ell\big(\theta_k'; \{(\mathbf{x}, \mathbf{y})\}\big)$$
$$\approx \ell\big(\theta_k; \{(\mathbf{x}, \mathbf{y})\}\big) - \ell\big(\theta_k - \alpha g_k; \{(\mathbf{x}, \mathbf{y})\}\big) \qquad \text{(Last layer update only)}$$
$$\approx \ell\big(\theta_k; \{(\mathbf{x}, \mathbf{y})\}\big) - \Big(\ell\big(\theta_k; \{(\mathbf{x}, \mathbf{y})\}\big) - \alpha g_k^\top \nabla_{\theta_k} \ell\big(\theta_k; \{(\mathbf{x}, \mathbf{y})\}\big)\Big) \quad \text{(Taylor approximation)}$$
$$= \alpha g_k^\top \nabla_{\theta_k} \ell\big(\theta_k; \{(\mathbf{x}, \mathbf{y})\}\big) = \alpha \|g_k\|_2^2.$$

| **Algorithm 1** Meta-training of GradNCP | **Algorithm 2** Meta-testing of GradNCP |
|---|---|
| **Input:** Initial $\theta_0$, $\{\mathbf{s}_i\}_{i=1}^N$, $\gamma, \alpha, \beta, \lambda, K, L$ | **Input:** Test signal $\mathbf{s}$, learned initialization $\theta_0$, $K_{\text{test}}$ |
| 1: **while** not converge **do** | 1: Extract context $\mathcal{C}_{\text{full}}$ from $\mathbf{s}$. |
| 2:     Sample batch $\{\mathbf{s}_1, \ldots, \mathbf{s}_B\}$. |     # Where typically $K_{\text{test}} > K + L$ |
| 3:     **for all** $b = 1$ to $B$ **do** | 2: **for all** $t = 0$ to $K_{\text{test}} - 1$ **do** |
| 4:         Extract context $\mathcal{C}_{\text{full}}$ from $\mathbf{s}_b$. | 3:     # Context pruning |
| 5:         **for all** $k = 0$ to $K - 1$ **do** | 4:     $\mathcal{C}_{\text{high}} = \text{TopK}(\mathcal{C}_{\text{full}}; R_t, \gamma)$ |
| 6:            # Online Context pruning |     # Compute gradient scaling |
| 7:            $\mathcal{C}_{\text{high}}^k = \text{TopK}(\mathcal{C}_{\text{full}}; R_k, \gamma)$ |     $g_t^{\text{test}} = \frac{\|\nabla_{\theta_t}\mathcal{L}_{\text{MSE}}(\theta_t, \mathcal{C}_{\text{high}})\|}{\|\nabla_{\theta_t}\mathcal{L}_{\text{MSE}}(\theta_t, \mathcal{C}_{\text{full}})\|}\nabla_{\theta_t}\mathcal{L}_{\text{MSE}}(\theta_t, \mathcal{C}_{\text{full}})$ |
| 8:            $\theta_{k+1} \leftarrow \theta_k - \alpha\nabla_{\theta_k}\mathcal{L}_{\text{MSE}}(\theta_k; \mathcal{C}_{\text{high}}^k)$ |     # Adaptation with full context |
| 9:         **end for** | 5:     $\theta_{t+1} \leftarrow \theta_t - \alpha \cdot g_t^{\text{test}}$ |
| 10:         # Generate target in L steps | 6: **end for** |
| 11:         $\theta_{K+1}^{\text{boot}} \leftarrow \theta_K - \alpha\nabla_{\theta_K}\mathcal{L}_{\text{MSE}}(\theta_K; \mathcal{C}_{\text{full}})$ | **Output:** $\mathcal{L}_{\text{MSE}}(\theta_T, \mathcal{C}_{\text{full}}), \theta_T$ |
| 12:         $\ldots$ | |
| 13:         $\mathcal{L}_{\text{total}}^b = \mathcal{L}_{\text{MSE}}(\theta_K; \mathcal{C}_{\text{full}}) + \lambda\mu(\theta_K, \theta_{K+L}^{\text{boot}})$ | |
| 14:     **end for** | |
| 15:     $\theta_0 \leftarrow \theta_0 - \beta\frac{1}{B}\sum_{b=1}^B \nabla_{\theta_0}\mathcal{L}_{\text{total}}^b$ | |
| 16: **end while** | |
| **Output:** $\theta_0$ | |

where the equivalence in the last layer results from all gradients w.r.t $\theta_k^{\text{base}}$ in $\nabla_{\theta_k}\ell\big(\theta_k; \{(\mathbf{x}, \mathbf{y})\}\big)$ being multiplied with zero by their corresponding entry in $g_k$. As the gradient for the final layer takes the standard form for a linear model (see non-linear version and the full derivation in Appendix B), we can define our sample importance score $R_k^{\text{GradNCP}}$ under $\mathcal{L}_{\text{MSE}}$ at adaptation step $k$ as:

$$R_k^{\text{GradNCP}}(\mathbf{x}, \mathbf{y}) := \left\| \big(\mathbf{y} - f_{\theta_k}(\mathbf{x})\big)\big[\phi_{\theta_k^{\text{base}}}(\mathbf{x}), \mathbf{1}\big]^\top \right\| \tag{2}$$

where $\|\cdot\|$ is the Frobenius norm and $\phi_{\theta_k^{\text{base}}}(\cdot)$ the penultimate feature vector, such that $f_{\theta_k}(\mathbf{x}) := W_k\phi_{\theta_k^{\text{base}}}(\mathbf{x}) + b_k$. This score only requires a single forward pass (and an additional small matrix multiplication) without the need for full backpropagation, making it straightforward and scalable to use for high-resolution signals. This is particularly suitable for an extensions of MAML which we use in practice [35], that considers the step size at each iteration $K$ of the inner learning problem part of the meta-optimization parameters, adding increased flexibility in terms of both choosing the most suitable data and update size for each iteration. We further emphasize that while motivated in the NF context, this approximation can be generally applicable in curriculum learning settings [44, 58].

## 2.3 Reducing Bias and Myopia through Bootstrap Correction

Despite the careful selection of $\mathcal{C}_{\text{high}}^k$, a large reduction in the size of the context set may inevitably introduce information loss. To mitigate this issue, we suggest counteracting any bias by reducing a distance to a parameter state obtained using optimization carried with the *full context set*. Specifically, after adapting the meta-learner for $K$ steps with the pruned context set, we generate a bootstrapped target $\theta_{K+L}^{\text{boot}}$ through optimization for $L$ additional steps starting from $\theta_K$. Crucially, we consider meta-optimization to be oblivious to the target generating process, i.e., do not backpropagate through its updates, saving memory otherwise required for second-order optimization. In automated differentiation frameworks, this can be implemented with a stop-gradient operation on $\theta_{K+L}^{\text{boot}}$, leading to each of the $L$ additional steps being significantly cheaper as well as improved target performance.

Given both sets of parameters, the desired correction effect can be achieved by regularizing $\theta_K$ to be close to the bootstrap target $\theta_{K+L}^{\text{boot}}$ according to a distance function $\mu(\theta_K, \theta_{K+L}^{\text{boot}})$ with $\ell_2$ distance being a suitable choice. This correction has two effects: Firstly, it forces increased plasticity in the initial $K$ steps as the only means to reduce the regularization loss to a target obtained using both additional information (i.e., access $\mathcal{C}^{\text{full}}$) and further optimization. This is possible as information about future learning dynamics is infused in the initial steps of learning through bootstrapping. In effect, the meta-learner teaches to improve itself. Secondly, as noted in the related work [17], bootstrapping reduces the phenomenon of myopia in meta-learning - a well-known issue significantly reducing the effectiveness of longer optimization loops at meta-test time (when no second-order optimization is needed and thus longer adaption procedures are possible). Flennerhag et al. [17] show that bootstrapping for this purpose leads to *guaranteed* performance improvements.

**Overall meta-learning objective.** In practice, we find that it is useful to calculate the final loss as a combination of the target correction term and the performance of $\theta_K$ *on the full context set*. For a given hyper-parameter $\lambda > 0$, the meta-objective of GradNCP becomes:

$$\mathcal{L}_{\texttt{total}}(\theta_0; \mathcal{C}_{\texttt{full}}) \coloneqq \mathcal{L}_{\texttt{MSE}}(\theta_K; \mathcal{C}_{\texttt{full}}) + \lambda\mu(\theta_K, \theta^{\texttt{boot}}_{K+L}), \tag{3}$$

a choice inspired by the common meta-learning practice of optimizing the meta-loss w.r.t the inner validation set performance, infusing further information into the initial $K$ adaptation steps. The full meta-training and -testing procedures are shown in Algorithms 1 & 2.

**Connection with target model regularization frameworks.** Addressing the issue of information loss through a target model can be viewed as an application of prior methods that engage the target model in instructing the meta-learner. While prior works aim to adapt a better target by further optimization [17, 31], or adapting from a better initialization [59], we believe such methods can be further improved by utilizing the full context set. Importantly, to the best of our knowledge, this work is the first attempt of introducing bootstrapping as a means to correct the effects of data pruning.

## 2.4 Gradient Re-scaling enables Effective Meta-test Optimization

Relieved from the memory requirements of second-order optimization and the burden of meta-learning myopia, the use of the full context set at meta-test time appears a natural choice. The naive combination of these ideas may not lead to desired results however, as the norm of the inner-loop gradient significantly deviates between meta-training and testing by construction, i.e., $\|\nabla_\theta \mathcal{L}_{\texttt{MSE}}(\theta; \mathcal{C}_{\texttt{high}})\| > \|\nabla_\theta \mathcal{L}_{\texttt{MSE}}(\theta; \mathcal{C}_{\texttt{full}})\|$, in accordance with the importance score $R^{\text{GradNCP}}_k$. Accordingly, we use a simple correction that ensures test-time gradient statistics are on the same order as statistics optimized for during meta-training. At step meta-test step $t$ we correct as:

$$g^{\texttt{test}}_t \leftarrow \frac{\|\nabla_{\theta_t} \mathcal{L}_{\texttt{MSE}}(\theta_t; \mathcal{C}_{\texttt{high}})\|}{\|\nabla_{\theta_t} \mathcal{L}_{\texttt{MSE}}(\theta_t; \mathcal{C}_{\texttt{full}})\|} \nabla_{\theta_t} \mathcal{L}_{\texttt{MSE}}(\theta_t; \mathcal{C}_{\texttt{full}}), \tag{4}$$

and then update $\theta_t$ with $g^{\texttt{test}}_t$ instead of $\nabla_{\theta_t} \mathcal{L}_{\texttt{MSE}}(\theta_t; \mathcal{C}_{\texttt{full}})$. We observe this to be a simple and sufficient means to drastically improving test-time performance when using $\mathcal{C}_{\texttt{full}}$.

## 3  Experiments

We now provide an empirical evaluation of GradNCP, systematically verifying claims made throughout the manuscript and thus supporting the suitability of its constituent components. At this point, we remind the reader of the key questions raised thus far:

1. **Efficiency and context scoring.** Can GradNCP's online context selection mechanism reduce memory and compute requirements vis-a-vis standard meta-learning? If so, what is the quality of the introduced scoring function $R^{\text{GradNCP}}_k$ and how can its approximations be justified? Finally, do selected context points correspond to humanly interpretable features?

2. **Bootstrap correction and myopia.** What is the effect of bootstrap correction on any irreducible error introduced through context pruning? Do we observe improvements on the issue of myopia? How do we know that bootstrapping provides a well-conditioned target model?

3. **Overall evaluation.** How does GradNCP compare to state-of-the-art meta-learning techniques for NFs? What are the necessary conditions for the highest-quality results at test time? Do introduced efficiency techniques enable learning on more complex signals?

Before answering each question, we first lay out the experimental protocol used throughout:

**Baselines.** We compare GradNCP with existing meta-learning schemes for NFs, including the MAML [15] (known as Learnit [61] in the NF literature), transformer-based meta-learning (TransINR; [6]) and the recent Instant Pattern Composer (IPC; [29]). Where appropriate, we also show the effect of *tablua rasa* optimization from a random initialization (Random Init.) to highlight the need for accelerated learning. Presented TransINR and MAML/Learnit results have been obtained with reference to the official implementation (and recommended hyperparameters for TransINR) while we adjust MAML/Learnit's number of inner gradient steps for a fair comparison with GradNCP. IPC values correspond to results directly published by the authors.

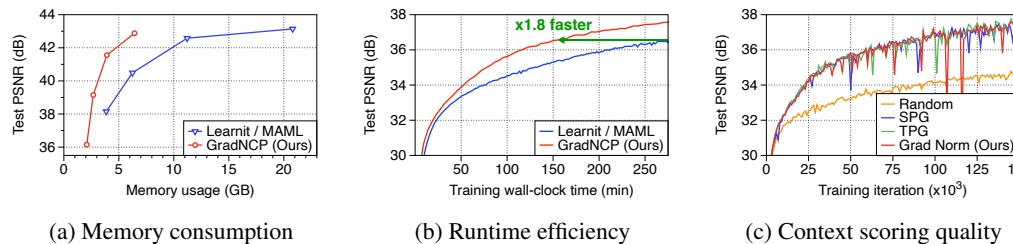

|  (a) Memory consumption | (b) Runtime efficiency | (c) Context scoring quality |

Figure 3: Comparing the learning efficiency of GradNCP to standard meta-learning. (a) GradNCP drastically decreases memory usage when varying $K$ at meta-training time. (b) GradNCP achieves the same performance in 55% of wall-clock time ($1.8\times$ speedup). (c) GradNCP scoring maintains equivalent performance while reducing runtime complexity: $\mathcal{O}(M^2)$ to $\mathcal{O}(M)$ relative to self-prediction gain (SPG) and target-prediction gain (TPG) [21] approximated with last layer update.

**NF architectures.** We use SIREN [54] as the base architecture for all experiments, and additionally consider NeRV [5] in the video domain. We provide more comparisons on other architectures, such as Fourier feature networks (FFN) [60] in Appendix D.3.

**Evaluation setup.** To ensure a fair comparison, we control for the number of adaptation steps and/or memory requirements when comparing Learnit/MAML with GradNCP. Quantitative results are reported using Peak Signal-to-Noise Ratio (PSNR; higher is better) as well as perceptual similarity for image and video datasets by using LPIPS (lower is better) [69] and SSIM (higher is better) [65]. For video datasets, we evaluate these metrics in a frame-wise manner and average over the whole video following NeRV[5]. All ablation studies presented show results on CelebA ($178 \times 178$) using $K = 16$ inner gradient steps, $L = 5$ bootstrap target construction steps and $\gamma = 0.25$ for GradNCP. The MAML / Learnit configuration with equivalent memory use corresponds to $K = 4$ while both methods use $K_{\text{test}} = 16$ test time adaptation steps unless otherwise stated.

**Datasets.** Highlighting the versatility of NFs, we consider four different modalities, namely images, videos, audio, and manifolds. For images, we follow Learnit and use CelebA [36], Imagenette [23], and pictures including Text [61] while additionally considering the high-resolution fine-grained datasets CelebA-HQ [26] and AFHQ [7]. To test under high visual diversity we also use all 1,000 classes of ImageNet [8]. Pushing memory requirements past current limits, we consider the video dataset UCF-101 [57] using resolutions ($128\times128\times16$, $256\times256\times32$) to demonstrate the scalability of GradNCP. Finally, we also test on audio using the speech dataset Librispeech [42] and on manifolds by considering climate data on the globe using ERA5 [22].

## 3.1 Online Context Pruning leads to Highly Efficient Meta-Learning

First turning to the primary motivation for our work, we show a comparison in terms of memory efficiency and runtime in Figures 3a and 3b, noting that GradNCP leads to both a significant improvement in the memory/performance trade-off. This enables either significant runtime speedups on the same machine or meta-learning on significantly higher dimensional signals (as we will show later). Given such encouraging results, how do we know whether approximations made for $R_k^{\text{GradNCP}}$ in Section 2.2 are justifiable? We attempt to answer this in Figure 3c by comparing to existing curriculum-learning metrics, noting (i) that approximating the effect of a full network update using the last layer only continues to consistently outperform random sub-sampling (ii) measuring the learning effect of a context point by the reduction in error on itself does not decrease performance (showing equivalent performance of SPG and TPG) and (iii) that our proposed first-order Taylor series approximation does successfully alleviates the need for an update without performance degradation.

What explains the robustness of both approximations? In Figure 4 we remarkably notice that when used in a meta-learning setting, the correlations between the last layer and full network gradient consistently improve with the number of inner adaptation steps (unlike for a random initialization). Remarkably, this suggests that meta-learning dynamics *push parameters to a state where the approximation assumptions above are satisfied* while simultaneously improving performance. Moreover, as we mentioned in Figure 1, a visualization of the context set $\mathcal{C}_{\text{high}}^k$ generally corresponds to interpretable features, with many more examples of this phenomenon shown in Appendix D.1.

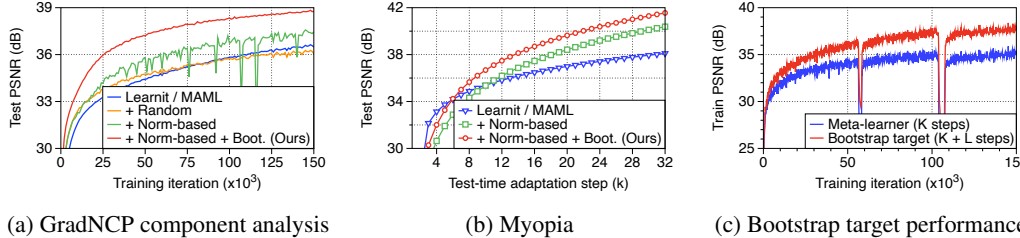

| (a) GradNCP component analysis | (b) Myopia | (c) Bootstrap target performance |

Figure 5: Analysing the effects of bootstrapping. (a) Bootstrapping reduces error introduced by context pruning and stabilizes meta-training. (b) Bootstrap correction allows learning past the meta-training horizon ($K = 16$), reducing myopia. (c) The bootstrap model achieves consistently higher performance, providing a well-conditioned target.

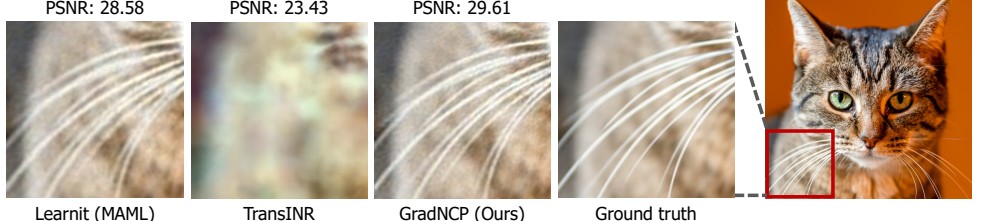

| Learnit (MAML) | TransINR | GradNCP (Ours) | Ground truth |

Figure 6: Qualitative comparison between GradNCP and baselines on AFHQ ($512 \times 512$).

## 3.2 Understanding the Effects of Bootstrap Correction

Given the high quality of our online-context point selection scheme, what additional contributions are made by the bootstrap correction (Section 2.3)?

We answer this question in Figure 5a, noting that bootstrapping indeed leads to consistent performance improvements while also stabilizing training. Note that this effect is distinct from any reduction in myopia (as we keep $K = K_{\text{test}}$ fixed). Myopia nevertheless exists, as we show in Figure 5b, noting that both Learnit / MAML and GradNCP w/o bootstrap correction result in reduced test-time adaptation results. It is noteworthy that the rate of improvement for each additional step drastically decreases for Learnit / MAML, showing the reduced plasticity effect we discussed throughout the manuscript.

One possible concern for bootstrapping in our setting is that the GradNCP loss (3) could be trivially minimized by finding a local minima that cannot be escaped in the $L$ additional steps afforded to the target model, thus leading to a low regularization loss but also removing any positive effect of bootstrapping. Fortunately, this is not the case, as we show in Figure 5c, clearly highlighting that the $K$ step meta-learner consistently trails an improved target model, providing a well-conditioned regularization target.

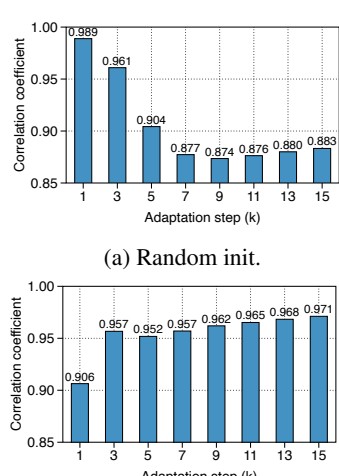

(a) Random init.

(b) Meta-learned init.

Figure 4: Meta-learning dynamics improve approximation quality over time. Shown is the correlation between the last layer and the entire network gradient.

## 3.3 GradNCP achieves State-of-the-Art Results in Meta-Learning Neural Fields

Having motivated and verified GradNCP's components, we now turn to our main results, presenting results for images in Table 1, videos in Table 2, as well as audio and manifolds in Table 3. Overall, GradNCP significantly and consistently outperforms state-of-the-art meta-learning methods by a large margin, leading to high-quality NFs as shown in Figure 6, exhibiting clear superiority over the baselines in capturing high-frequency components - a direct benefit of online context pruning, due to additional learning capacity dedicated to such high-frequency details in later adaptation steps. We provide more qualitative results in Appendix D.11.

Table 1: Reconstruction performance of SIREN on image datasets of various resolutions. We report results as PSNR [dB] (↑) / SSIM (↑) / LPIPS (↓) using the same number of adaptation steps for all schemes. OOM denotes the out-of-memory on a single NVIDIA A100 40GB GPU. Here IPC indicates Instance Pattern Composers with results directly taken from [29].

| | **CelebA** (178 × 178) | **Imagenette** (178 × 178) | **Text** (178 × 178) |
|---|---|---|---|
| Random Init. | 19.94 / 0.532 / 0.708 | 18.57 / 0.443 / 0.810 | 15.37 / 0.574 / 0.755 |
| TransINR [6] | 32.37 / 0.913 / 0.068 | 28.58 / 0.850 / 0.165 | 22.70 / 0.898 / 0.085 |
| IPC [29] | 35.93 / - / - | 38.46 / - / - | - / - / - |
| Learnit / MAML [61] | 38.28 / 0.964 / 0.010 | 35.66 / 0.950 / 0.014 | 30.31 / 0.956 / 0.018 |
| **GradNCP (Ours)** | **40.60 / 0.976 / 0.005** | **38.72 / 0.972 / 0.005** | **32.33 / 0.976 / 0.007** |

| | **ImageNet** (256 × 256) | **AFHQ** (512 × 512) | **CelebA-HQ** (1024 × 1024) |
|---|---|---|---|
| Random Init. | 18.72 / 0.434 / 0.839 | 18.57 / 0.488 / 0.856 | 12.21 / 0.574 / 0.820 |
| TransINR [6] | 28.01 / 0.818 / 0.199 | 23.43 / 0.592 / 0.573 | ——— OOM ——— |
| Learnit / MAML [61] | 31.44 / 0.887 / 0.100 | 28.58 / 0.751 / 0.354 | 27.66 / 0.781 / 0.513 |
| **GradNCP (Ours)** | **32.52 / 0.898 / 0.068** | **29.61 / 0.786 / 0.286** | **28.90 / 0.789 / 0.438** |

Table 2: Reconstruction performance of meta-learned SIREN and NeRV. We consider two different resolutions of the UCF-101 dataset. OOM denotes unavailable results due to running out-of-memory on a single NVIDIA A100 40GB GPU.

| Resolution | Network | Method | PSNR (↑) | SSIM (↑) | LPIPS (↓) |
|---|---|---|---|---|---|
| 128×128×16 | SIREN | TransINR [6] | 15.14 | 0.360 | 0.636 |
| | | Learnit / MAML [61] | 25.46 | 0.720 | 0.363 |
| | | **GradNCP (Ours)** | **26.92** | **0.781** | **0.223** |
| | NeRV | Learnit (MAML) [61] | 28.86 | 0.871 | 0.140 |
| | | **GradNCP (Ours)** | **35.28** | **0.959** | **0.015** |
| 256×256×32 | SIREN | TransINR [6] | ——— OOM ——— | | |
| | | Learnit / MAML [61] | ——— OOM ——— | | |
| | | **GradNCP (Ours)** | **22.92** | **0.640** | **0.521** |
| | NeRV | Learnit / MAML [61] | 23.75 | 0.659 | 0.422 |
| | | **GradNCP (Ours)** | **28.65** | **0.842** | **0.201** |

Table 3: PSNR of meta-learned SIREN on (a) Librispeech and (b) ERA5 (181×360).

(a) Librispeech

| | PSNR (↑) | |
|---|---|---|
| Method | 1 sec | 3 sec |
| TransINR [6] | 39.22 | 33.17 |
| IPC [29] | 40.11 | 35.38 |
| Learnit / MAML [61] | 39.55 | 31.39 |
| **GradNCP (Ours)** | **43.25** | **36.24** |

(b) ERA5

| Method | PSNR (↑) |
|---|---|
| Learnit / MAML [61] | 64.91 |
| **GradNCP (Ours)** | **75.11** |

We remark that one of the nice properties of GradNCP is its modality- and model-agnosticism. On the other hand, we find TransINR struggles to generalize for some modalities and architectures: namely it requires (a) tokenization, which is not straightforward to apply for spherical coordinate datasets (e.g., ERA5), and may require the framework to deal with sequences of tokens that are too long (e.g., videos), and (b) notable modifications to the architecture when it consists of non-MLP layers (e.g., NeRV) as the framework is specific to MLPs.

**High-resolution signals.** One of the significant advantages of GradNCP is its exceptional memory efficiency, which allows us to meta-learn on large-scale high-resolution signals. As demonstrated in Table 1 and Table 2, GradNCP can even be trained on 256×256×32 resolution videos or 1024×1024 resolution images, a scale previously not possible for prior work (even under the constraint of NVIDIA A100 40GB GPU) due to their intensive memory usage (see Figure 3a).

**GradNCP for generalization tasks.** To also verify that GradNCP can be used for generalization task where the meta-learning context and target sets are disjoint. To this end, we apply GradNCP for the NeRF [40] experiment on ShapeNet Car [4] to verify the generalization performance of GradNCP. Here, following the setup in Tancik et al. [61], we train NeRFs with a given single view of the 3D scene and evaluate the performance on other unseen views. As shown in Table 4, GradNCP significantly outperforms other baselines including Learnit / MAML (e.g., 22.80 → 24.06 dB), and even achieves better performance than TransINR which utilizes an additional (large) Transformer. We believe using GradNCP for large-scale generalization of NFs (e.g., NeRF for Google street view) will be an interesting future direction to explore, where we believe the efficiency of GradNCP can contribute to this field.

Table 4: PSNR (dB) of unseen view synthesis of NeRF on ShapeNet Cars. We adapt NeRF with a single view of a 3D scene then evaluate the quality of unseen views.

| Method | PSNR |
|---|---|
| Learnit [61] | 22.80 |
| TransINR [6] | 23.78 |
| **GradNCP (Ours)** | **24.06** |

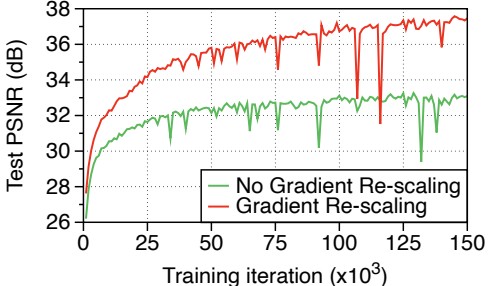

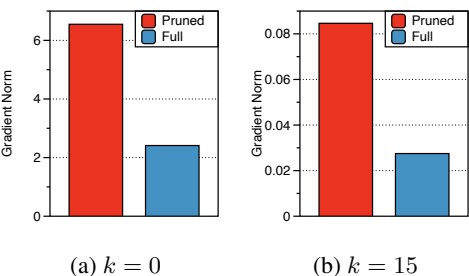

(a) $k = 0$                                (b) $k = 15$

Figure 7: Effect of meta-test time gradient re-scaling. We apply re-scaling when adapting with full context set on SIREN trained with GradNCP.

Figure 8: Gradient norm of the full context set $\mathcal{C}_{\texttt{full}}$ and the gradient norm-based pruned context set $\mathcal{C}_{\texttt{high}}^k$ at iteration $k$.

Table 5: Effect of context set selection ratio $\gamma$ on SIREN meta-learned with CelebA (178 ×178). $\gamma$ controls the performance and the memory trade-off when using GradNCP.

| Selection ratio ($\gamma$) | 0.05 | 0.10 | 0.15 | 0.20 | 0.25 | 0.5 | 0.75 | 1.0 |
|---|---|---|---|---|---|---|---|---|
| PSNR (dB) | 31.20 | 35.00 | 36.67 | 37.59 | 38.90 | 39.25 | 39.70 | 39.98 |
| Memory (GB) | 1.93 | 2.41 | 3.05 | 3.45 | 3.91 | 6.31 | 9.22 | 11.04 |

### 3.4 Additional Analysis

**Gradient re-scaling analysis.** Here, we investigate the importance of gradient re-scaling. To show the effect of gradient re-scaling at the meta-test time, we track the test PSNR during training SIREN with GradNCP (that does not use the Bootstrapped correction) on CelebA (178×178). Here, we report the PSNR when the meta-initialization is adapted with the full context set on the meta-test stage. As shown in Figure 7, the gradient re-scaling significantly improves the performance.

To further investigate the necessity of gradient re-scaling, we additionally analyze the gradient norm during the (inner loop) adaptation phase of meta-training. Here, we plot the gradient norms at the adaptation step $k$ measured by the full context set $\mathcal{C}_{\texttt{full}}$, and the gradient norm-based pruned context set $\mathcal{C}_{\texttt{high}}^k$, respectively. As shown in Figure 8, our results indicate that the norm of the gradients exhibits significant variations, with some steps exhibiting $3\times$ larger gradients when using the pruned context set. This suggests that the meta-learner employed a larger step size during training, which highlights the importance of test-time gradient re-scaling. We also find that re-scaling the gradient with the loss ratio of the full context set and the sampled context leads to similar performance improvements and may serve as a faster alternative, as it eliminates the need for gradient calculation twice, i.e., showing 40.53 in PSNR where the gradient re-scaling with gradient norms shows 40.80.

**Selection ratio analysis.** We also have investigated the tradeoff between performance and memory requirements of GradNCP by varying the value of the context set selection ratio $\gamma$ in our meta-training process on CelebA (178×178). As shown in Table 5, the memory usage decreases proportionally as one decreases the selection ratio, while the performance is maintained fairly well even for relatively small values of $\gamma = 0.15$, i.e., 36.67 (dB). For the main experiment, we mainly use $\gamma = 0.25$, which shows high performance while significantly reducing the memory usage, e.g., 75% of memory.

## 4 Discussion and Conclusion

We propose GradNCP, an efficient and effective method for fast and scalable neural field learning through online-pruning of the context set at each meta-training iteration, significantly reducing memory usage. We demonstrate that GradNCP improves performance over various modalities and, more importantly, exhibits superior memory efficiency allowing it to be the first method capable of meta-learning on exceptionally high-resolution signals.

**Future work and limitation.** We believe extending GradNCP to extreme high-resolution signals (e.g., a long 8K video), where even a single forward pass is not possible, would be an interesting future direction to explore. We believe that a great variety of techniques can be developed in this direction, e.g., iterative tree-search of high gradient norm samples by starting from a low-dimension grid and incrementally increasing the resolution of the sampled area.

## Acknowledgements

We thank Kyuyoung Kim and Jongjin Park for providing helpful feedbacks and suggestions in preparing an earlier version of the manuscript. This work was supported by Institute of Information & communications Technology Planning & Evaluation (IITP) grant funded by the Korea government (MSIT) (No.2019-0-00075, Artificial Intelligence Graduate School Program (KAIST), and No.2022-0-00713, Meta-learning applicable to real-world problems) and National Research Foundation of Korea(NRF) grant funded by the Korea government(MSIT) (No.RS-2023-00213710, Neural Network Optimization with Minimal Optimization Costs).

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

# Appendix

## A Experimental Details

In this section, we describe the experimental details of Section 3, including GradNCP and the baselines. We also provide the implementation of GradNCP at https://github.com/jihoontack/GradNCP.

### A.1 Dataset Details

**CelebA.** CelebA is a fine-grained dataset that consists of the face image of celebrities [36]. The dataset comprises 202,599 images, where we use 162K for training, 20K for validation, and 20K for testing. We resize the image into 178 and then center cropped of 178, which ends with a resolution of $178 \times 178$. We pre-process pixel coordinates into $[-1, 1]^2$ and signal values ranging from 0 to 1.

**Imagenette.** Imagenette is a 10-class subset of the ImageNet [8] dataset, which is comprised of 9,000 training and 4,000 test images [23]. We resize the image into 178 and then apply a center crop of 178, which ends with a resolution of $178 \times 178$. We pre-process pixel coordinates into $[-1, 1]^2$ and signal values ranging from 0 to 1.

**Text.** Text dataset consists of text image with a resolution of $178 \times 178$ [61]. We pre-process the pixel coordinates into $[-1, 1]^2$ and signal values ranging from 0 to 1.

**ImageNet.** ImageNet dataset is a large-scale natural image dataset consisting of 1,000 classes [8]. We use 1,281K images for training and evaluated the methods on 50K test images (known as the validation set of ImageNet). For image pre-processing, we resize the image into 256, then center cropped the image to get $256 \times 256$ resolution image. For coordinate pre-processing, we use pixel coordinates of $[-1, 1]^2$ and signal values ranging from 0 to 1.

**CelebA-HQ.** CelebA-HQ is a high-resolution fine-grained dataset, which includes images of celebrities [26]. We divided the dataset into 27,000 training and 3,000 test samples and pre-processed the pixel coordinates into $[-1, 1]^2$ and signal values ranging from 0 to 1. We consider two resolutions, i.e., $512 \times 512$ and $1024 \times 1024$.

**AFHQ.** AFHQ is a high-resolution fine-grained dataset, which includes animal faces consisting of 15,000 images at $512 \times 512$ resolution [7]. We divided the dataset into 14,336 training and 1,467 testing points and pre-processed the pixel coordinates into $[-1, 1]^2$ and signal values from 0 to 1.

**UCF-101.** UCF-101 is a video dataset comprising 13,320 videos (9,357 training and 3,963 test videos) with a resolution of $320 \times 240$, where the action classification consists of 101 classes [57]. Each video clip is center-cropped to $240 \times 240$ and then resized into $128 \times 128$ and $256 \times 256$ with a video clip length of 16 and 32, respectively. We pre-process the pixel coordinates into $[-1, 1]^3$ and signal values ranging from 0 to 1.

**Kinetics-400.** For the cross-domain adaptation purpose in Appendix D.7, we use the mini-Kinetics-200 dataset [67], which consists of 200 categories with the most training samples from the Kinetics-400 dataset [28]. We center-crop the videos to the same height and width and then resize them into $128 \times 128$ resolution, and use the frame of 16 clips. We pre-process the pixel coordinates into $[-1, 1]^3$ and signal values ranging from 0 to 1.

**ERA5.** ERA5 is a dataset comprised of temperature observations on a global grid of equally spaced latitudes and longitudes [22]. By following [11], we use the grid resolution of $181 \times 360$ by resizing the grid. We interpret each time step as an independent signal, and the dataset comprises 9,676 training points and 2,420 test points. For the input pre-processing, given latitudes $\rho$ and longitudes $\varphi$ are transformed into 3D Cartesian coordinates $\mathbf{c} = (\cos\rho\cos\varphi, \cos\rho\sin\varphi, \sin\rho)$ where latitudes $\rho$ are equally spaced between $-\frac{\pi}{2}$ and $\frac{\pi}{2}$ and longitudes $\varphi$ are equally spaced between 0 and $\frac{2\pi(n-1)}{n}$ where $n$ the number of distinct values of longitude (360).

**LibriSpeech.** LibriSpeech is an English speech recording collection at a 16kHz sampling rate [42]. By following Dupont et al. [12], we use the train-clean-100 split for training and test-clean split for testing, i.e., 28,539 training and 2,620 test samples. For the main experiments, we use the first 1 second and 3 seconds of each example (1 second contains 16,000 coordinates). For the pre-processing, we scale the coordinates into $[-50, 50]$.

### A.2 Training and Evaluation Details

**Network architecture details.** For the main experiment, we mainly use SIREN, a multi-layer perception (MLP) with sinusoidal activation functions [54], i.e., $\mathbf{x} \mapsto \sin(\omega_0(W\mathbf{x} + b))$ where $W, b$ are weight and biases of the MLP layer and $\omega_0$ is the fixed hyperparameter. For image, audio, and manifold datasets, we use SIREN with 5 layers with 256 hidden dimensions, and for video, we use 7 layers with the same hidden dimension. We used $\omega_0 = 50$ for the manifold dataset and use $\omega_0 = 30$ for the rest. We additionally consider NeRV [5] for the video dataset. For the UCF-101 dataset of $128 \times 128 \times 16$, we use 4 NeRV blocks, and for $256 \times 256 \times 32$, we use 5 NeRV blocks. Finally, we consider the Fourier feature network (FFN) in Appendix D.3, where we use the same network size as SIREN and the use Fourier feature scale $\sigma = 10$.

**Training details.** For all dataset, we use Adam optimizer [32] for the outer loop. We use the outer step of 150,000, except for learning the ImageNet dataset where we use 500,000 steps. For SIREN, we use the outer learning rate of $\beta = 3.0 \times 10^{-6}$ for Librispeech and use $\beta = 1.0 \times 10^{-5}$ for the rest. For NeRV, we use the outer learning rate of $\beta = 1.0 \times 10^{-4}$. As for the inner loop learning rate, we use $\alpha = 1.0 \times 10^{-2}$, and $\alpha = 1.0 \times 10^{-1}$ for SIREN and NeRV, respectively. For inner step number $K$, GradNCP is trained on a longer horizon than Learnit by multiplying $1/\gamma$ (which uses the same memory usage). For Learnit, we mainly use $K = 4$ for the main table, where we use $K = 1$ for CelebA-HQ ($1024 \times 1024$), $K = 5$ on UCF-101 ($256 \times 256 \times 32$) on NeRV, and $K = 20$ on UCF-101 ($128 \times 128 \times 16$) on NeRV. We use the same batch size for Learnit and GradNCP to fairly use the memory where the size was selected differently across the dataset (e.g., under the given GPU memory budget).

**Hyperparameter details for GradNCP.** We find that the hyperparameter introduced by GradNCP is not sensitive across datasets and architectures. For the context set selection ratio $\gamma$, i.e., the ratio of retaining coordinates, we use $0.25$ for most of the dataset except for Librispeech and ERA5, where we used $0.5$ (as we do not need to prune the context much for these low-resolution signals), and $0.5, 0.2$ when training NeRV on UCF-101 on 128 and 256 resolution, respectively. We found that for most of the datasets, the performance does not significantly decrease until $\gamma = 0.2$ while significantly reducing the memory, i.e., about 5 times. For bootstrap target correction hyperparameters, we used $L = 5$, and $\lambda = 100$, where we believe tuning these hyperparameters will indeed improve the performance much more (we did not tune extensively).

**Evaluation details.** For the evaluation, to fairly compare with the baseline, we use the same test-time adaptation step for Learnit and GradNCP (e.g., for CelebA experiments we use $K_{\text{test}} = 16$) which is the same step number that is used by GradNCP on meta-training. We additionally compare the results of test-time adaptation of TransINR in Section D.8.

**NeRF details.** By following prior works [61, 6], we use the simplified NeRF model [40] which uses a single network rather two networks (coarse and fine), and do not use the view direction for the network. We use a MLP with 6 layers with 256 hidden dimension with ReLU activation. We follow the same training detail of Tancik et al. [61], except for inner loop steps where we use 16 steps (rather than 100 steps) and first random sample 512 rays then select the important 128 rays by using $\gamma = 0.25$ (same ray size as the prior works).

**Resource details.** For the main development, we mainly use Intel(R) Xeon(R) Gold 6226R CPU @ 2.90GHz and a single RTX 3090 24GB GPU, except for high-resolution signals including CelebA-HQ of $1024 \times 1024$ and UCF-101 of $256 \times 256 \times 32$, where we use AMD EPYC 7542 32 Core Processor and a single NVIDIA A100 SXM4 40GB.

### A.3 Baseline Details

In this section, we explain the meta-learning baselines we used for evaluating GradNCP. We note that we additionally compare with memory-efficient meta-learning schemes in Appendix D.5.

- **Learnit** [61] mainly uses the second-order gradient version of MAML [15] for learning NFs.
- **TransINR** [6] utilizes Transformer as a meta-learner to predict the NF parameter with a given context set, and additionally proposed a parameter-efficient NF architecture specialized for MLP.
- **IPC** [29] denotes instant pattern composer which is a modulation technique for meta-learning NFs. This modulation uses a low-rank parametrization of NF weights where the authors used TransINR as a meta-learning approach to train these modulations.

# B  Derivation of Our Data Importance Score

In this section, we prove our sample importance score (shown in the following equation) well approximates the updated loss value when adapting the network using the given sample.

$$R_k^{\text{GradNCP}}(\mathbf{x}, \mathbf{y}) := \left\| (\mathbf{y} - f_{\theta_k}(\mathbf{x})) \left[ \phi_{\theta_k^{\text{base}}}(\mathbf{x}), \mathbf{1} \right]^\top \right\|. \tag{5}$$

We first review the derivation in the main text. Let $\theta_k^{\text{base}}$ be the set of parameters excluding the final layer and $g_k = [\mathbf{0}, \nabla_{W_k}\ell, \nabla_{b_k}\ell]$ a padded gradient involving zeros for all but the last layer's weights $W_k$ and biases $b_k$, we start from the SPG metric:

$$
\begin{aligned}
& \ell\big(\theta_k; \{(\mathbf{x}, \mathbf{y})\}\big) - \ell\big(\theta_k'; \{(\mathbf{x}, \mathbf{y})\}\big) & \\
& \approx \ell\big(\theta_k; \{(\mathbf{x}, \mathbf{y})\}\big) - \ell\big(\theta_k - \alpha g_k; \{(\mathbf{x}, \mathbf{y})\}\big) & \text{(Last layer update only)} \\
& \approx \ell\big(\theta_k; \{(\mathbf{x}, \mathbf{y})\}\big) - \Big( \ell\big(\theta_k; \{(\mathbf{x}, \mathbf{y})\}\big) - \alpha g_k^\top \nabla_{\theta_k}\ell\big(\theta_k; \{(\mathbf{x}, \mathbf{y})\}\big) \Big) & \text{(Taylor approximation)} \\
& = \alpha g_k^\top \nabla_{\theta_k}\ell\big(\theta_k; \{(\mathbf{x}, \mathbf{y})\}\big) = \alpha \|g_k\|_2^2. &
\end{aligned}
$$

where $\alpha > 0$ is the step size of the optimization. We remove $\alpha$, as the sample importance ranking does not change as $\alpha$ is a positive value. Since NF training considers MSE loss for $\ell$, the gradient of the last layer weight and bias are as:

$$\nabla_{W_k}\ell\big(\theta_k; \{(\mathbf{x}, \mathbf{y})\}\big) = \nabla_{W_k} \left\| \Big( \mathbf{y} - \big( W_k \phi_{\theta_k^{\text{base}}}(\mathbf{x}) + b_k \big) \Big) \right\|_2^2 \tag{6}$$

$$= -2 \Big( \mathbf{y} - \big( W_k \phi_{\theta_k^{\text{base}}}(\mathbf{x}) + b_k \big) \Big) \phi_{\theta_k^{\text{base}}}(\mathbf{x})^\top \tag{7}$$

$$= -2 \big( \mathbf{y} - f_{\theta_k}(\mathbf{x}) \big) \phi_{\theta_k^{\text{base}}}(\mathbf{x})^\top \tag{8}$$

$$\nabla_{b_k}\ell\big(\theta_k; \{(\mathbf{x}, \mathbf{y})\}\big) = \nabla_{b_k} \left\| \Big( \mathbf{y} - \big( W_k \phi_{\theta_k^{\text{base}}}(\mathbf{x}) + b_k \big) \Big) \right\|_2^2 \tag{9}$$

$$= -2 \Big( \mathbf{y} - \big( W_k \phi_{\theta_k^{\text{base}}}(\mathbf{x}) + b_k \big) \Big) \tag{10}$$

$$= -2 \big( \mathbf{y} - f_{\theta_k}(\mathbf{x}) \big) \tag{11}$$

Given the gradient of the last layer in Eq. (8) and (11), the final score is derived as follows:

$$R_k^{\text{GradNCP}}(\mathbf{x}, \mathbf{y}) := \|g_k\|_2^2 = \left\| \left[ \nabla_{W_k}\ell\big(\theta_k; \{(\mathbf{x}, \mathbf{y})\}\big) \ \ \nabla_{b_k}\ell\big(\theta_k; \{(\mathbf{x}, \mathbf{y})\}\big) \right] \right\|_2^2 \tag{12}$$

$$= 2 \left\| (\mathbf{y} - f_{\theta_k}(\mathbf{x})) \left[ \phi_{\theta_k^{\text{base}}}(\mathbf{x}), \mathbf{1} \right]^\top \right\|. \tag{13}$$

Since the order (or the ranking) of the gradient norm value does not change by multiplying the positive value, we simply removed 2 from the final score. For NF with non-linear final layer (e.g., NeRV), one can easily derive the following GradNCP criteria: $\left\| (\mathbf{y} - f_{\theta_k}(\mathbf{x})) \sigma'\big( W_k \phi_{\theta_k^{\text{base}}}(\mathbf{x}) + b_k \big) \left[ \phi_{\theta_k^{\text{base}}}(\mathbf{x}), \mathbf{1} \right]^\top \right\|$ where the output of NF is $f_{\theta_k}(\mathbf{x}) = \sigma\big( W_k \phi_{\theta_k^{\text{base}}}(\mathbf{x}) + b_k \big)$, $\sigma(\cdot)$ is the element-wise non-linear operator (e.g., Sigmoid) and $\sigma'$ is the derivative of $\sigma$.

# C  Related Work

**Neural fields (NFs).** NFs, also referred to as implicit neural representations (INRs), have emerged as a new paradigm for representing complex, continuous signals [43, 60]. This approach encodes the signal with a neural network, typically a multi-layer perceptron (MLP) in conjunction with sinusoidal activations [54] or positional encoding [40]. They have become popular as their number of parameters do not strictly scale with the resolution of the signal [39, 52], easing modeling of multi-modal signals [9, 37] and showing potential for new approaches to prominent applications and downstream tasks, including data compression [10, 49], classification [11, 2], and generative modeling [55, 68]. However, fitting NFs is quite costly, especially for high-resolution signals [30]. To tackle this issue, we develop an efficient meta-learning framework for large-scale NF training which significantly reduce the memory usages of meta-learning.

**Efficient meta-learning.** There have been several works on developing (memory) efficient algorithms in the field of meta-learning. Typically, such algorithms have been explored in amortization-based (or encoder-based) schemes, e.g., Prototypical Networks [56, 3] and Neural Processes [20, 19, 18]. Unlike optimization-based meta-learning, however, these methods are somewhat limited in applicability to diverse modalities and NF architectures (requiring modality and model-specific design).

In the optimization-based regime, several memory efficient schemes have been introduced, including first-order MAML [15], Reptile [41], Implicit MAML [45] and continual trajectory shifting [51], that do not use the second-order gradient adaptation. However, recent works have shown that first-order optimization-based schemes can underperform for NFs [11, 12]. We also observed a similar result as summarized in Table 8 of Appendix D.5. Recently, network sparsity has been combined with meta-learning NFs [34, 48], which may help reduce computation at inference time but still requires some memory usage when meta-learning and more importantly can reduce the network expressive power. In this paper, we focus on developing a memory-efficient meta-learning framework that is built upon second-order gradient-based schemes for effective NF learning.

**Sparse data selection.** GradNCP is related to areas of machine learning which focus on identifying subsets of a dataset. For instance, data pruning focuses on designing data importance metrics to reduce the dataset without comprising the performance [63, 14, 27, 58], memory-based techniques of continual learning use subsets of past tasks to prevent catastrophic forgetting [62, 47], dataset distillation proposes the use of optimized few synthetic data for the same purpose [64], and active learning study way to identify and label data points to facilitate efficient learning progress during subsequent online updates [50, 13]. In this paper, we develop an efficient online context pruning scheme for meta-learning NFs by drawing inspiration from a recent work of curriculum learning to estimate the sample importance [21].

# D  More Experimental Results

## D.1  Gradient Norm Correlation

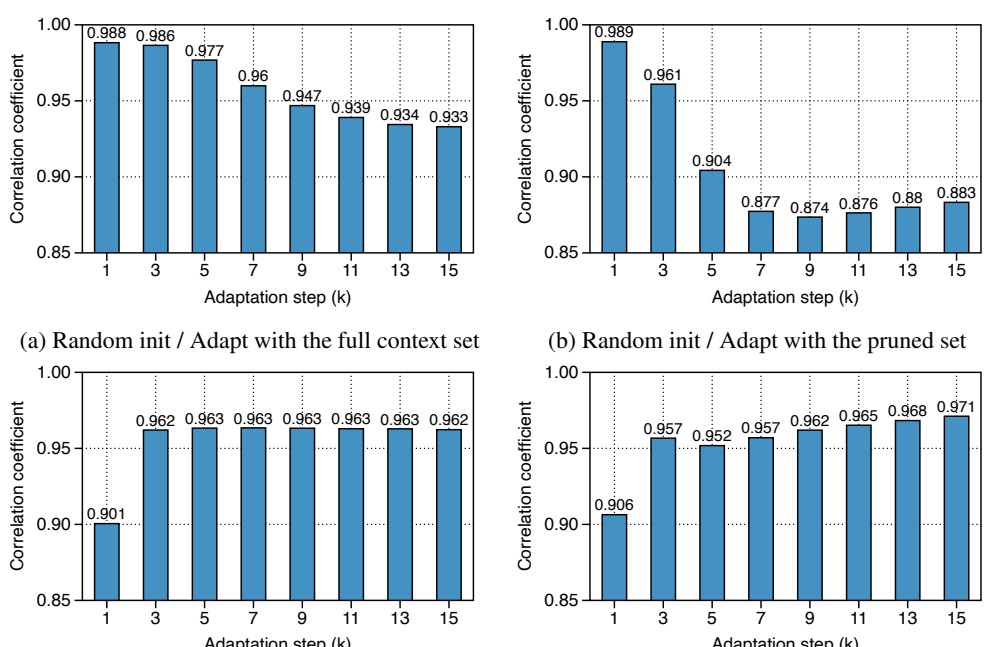

(a) Random init / Adapt with the full context set    (b) Random init / Adapt with the pruned set

(c) Meta-learned init / Adapt with the full context set    (d) Meta-learned init / Adapt with the pruned set

Figure 9: Correlation between the gradient norm of the entire network and the last layer network. We measure the Pearson correlation coefficient between the entire gradient norm and the last layer gradient norm on individual signals of the CelebA test dataset (i.e., correlation computed on $178 \times 178$ elements), then report the average correlation across signals. Here the pruned set indicates the context pruning with GradNCP.

In this section, we compare the correlation between the gradient norms of the entire network and the last layer to verify that our last layer gradient norm is a reasonable approximation. We first note that the entire network gradient norm is an approximation for the SPG score (i.e., by using the first-order Taylor series on SPG) without the last layer gradient approximation, hence, serves as a reasonable measure for the sample importance. It is worth noting that, while the entire gradient norm is a reasonable score, this is still computationally too heavy to use online, as it requires computing the gradient w.r.t to the entire network parameter with $O(M)$ time complexity where our last layer gradient norm only needs a forward pass of $O(M)$ (remark that $M$ is the context set size).

To this end, we measure the Pearson correlation coefficient between the entire gradient norm and the last layer gradient norm on individual signals of the CelebA test dataset (i.e., correlation computed on $178 \times 178$ elements), then report the average correlation across signals. As shown in Figure 9, first the entire gradient norm and the last layer gradient norm are both highly correlated in all cases indicating that our approximation using the last layer only optimization is quite reasonable. Furthermore, remarkably, when computed on the meta-learned initialization, the correlations between the last layer and the entire network gradient norm consistently improve during the inner adaptation steps (unlike for the random initialization). This indicates that the meta-learning dynamics push parameters to a state where the approximation assumptions above are satisfied while simultaneously improving performance.

## D.2 Ablation Study on Boostrapped Target Adaptation Steps

Table 6: PSNR (dB) of SIREN meta-learned on CelebA dataset ($178 \times 178$) when increasing the number of gradient steps ($L$) for generating bootstrapped target. Note that '0' indicates no bootstrapped correction (i.e., only use the gradient norm-based pruning over MAML).

| $L$ | 0 | 1 | 2 | 5 | 10 |
|---|---|---|---|---|---|
| PSNR (dB) | 37.49 | 37.78 | 38.20 | 38.90 | 38.99 |

We also perform an ablation study regarding the step for generating bootstrapped target $\theta^{\texttt{boot}}$. As shown in Table 6, we found that the performance rapidly improves until $L = 5$ and observed moderate improvement by further increasing $L$. During the development, we found that $L = 5$ is already quite effective across all modalities and datasets, and we believe tuning the hyperparameter will improve the performance.

## D.3 Comparison with Loss-based Selection

Table 7: Comparison of data selection criteria. We report the PSNR (dB) of meta-learned neural fields across various modalities when using loss-based (Loss) and gradient norm-based (Grad) top-K data selection. Bold indicates the best result of each group.

(a) Bounded activation

| | | | | | | SIREN | | | | |
|---|---|---|---|---|---|---|---|---|---|---|
| | CelebA | Imagenette | Text | AFHQ | CelebA-HQ | UCF-101 (128) | UCF-101 (256) | Libri (1sec) | Libri (3sec) | ERA5 |
| Loss | 40.54 | 37.71 | **33.11** | 29.37 | 28.89 | 26.59 | 22.76 | **43.40** | **36.45** | 74.10 |
| Grad | **40.60** | **38.72** | 32.33 | **29.61** | **28.90** | **26.92** | **22.92** | 43.25 | 36.24 | **75.11** |

(b) Non-bounded activation

| | NeRV | | FFN | | | |
|---|---|---|---|---|---|---|
| | UCF-101 (128) | UCF-101 (256) | CelebA | Text | Libri (1sec) | Libri (3sec) |
| Loss | 33.99 | 28.58 | 28.81 | 21.93 | 33.78 | 31.56 |
| Grad | **35.28** | **28.65** | **29.36** | **22.52** | **33.93** | **32.04** |

During the initial development, we also tried the loss-based (or error-based) selection [44], showing a good performance as well, where GradNCP consistently exhibited better performance (see Table 7, e.g., $33.99 \rightarrow 35.28$ dB in UCF-101). We believe this is because the loss-based selection is an approximation of GradNCP and thus GradNCP can provide better data pruning results. Specifically,

assuming that the penultimate feature has a bounded norm, i.e., $\|\phi(\cdot)\| \leq C$ where $C \in \mathbb{R}^+$, the GradNCP score can be approximated to loss-based selection as follows: $\|(\mathbf{y} - f(\mathbf{x}))[\phi(\mathbf{x})\ \mathbf{1}]^\top \leq \|\mathbf{y} - f(\mathbf{x})\|\|[\phi(\mathbf{x})\ \mathbf{1}]\| \leq C\|\mathbf{y} - f(\mathbf{x})\|$. Since some setups consider SIREN (i.e., sine activation), the assumption holds for these setups, thereby showing similar performance. While this condition does not always hold for general NF architectures that use ReLU activations (e.g., NeRV and FFN), thereby showing a consistent improvement than loss-based selection under NF architectures with ReLU activations. Therefore, we emphasize that our gradient-based selection can provide more accurate results than the loss-based selection, regardless of NF architectures.

## D.4 Last Layer Gradient Norm Statistic of Coordinates

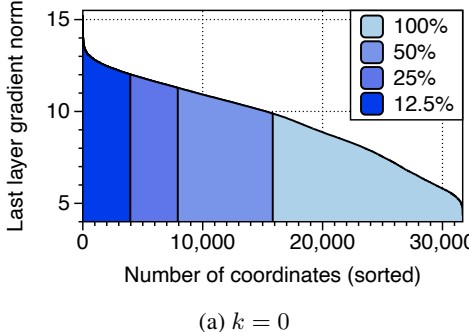
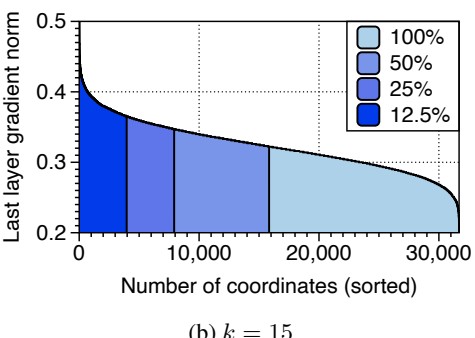

(a) $k = 0$          (b) $k = 15$

Figure 10: Last layer gradient norm statistic of coordinates where the indexes are sorted with the loss value and the highlighted region indicates the pruned context set (with gradient norm-based) by a given sampling ratio hyper-parameter $\gamma \times 100$ (%). The loss is measured on the meta-training set. $k$ indicates the adaptation step number. We meta-learn SIREN on CelebA ($178 \times 178$) with GradNCP.

To understand the behavior of GradNCP, we analyze the gradient norm statistics of the coordinates. Here, we visualize the last layer gradient norm of the given context set $\mathcal{C}_{\texttt{full}}$ at adaptation iteration $k$, namely $\{R_k(\mathbf{x}, \mathbf{y}) | (\mathbf{x}, \mathbf{y}) \in \mathcal{C}_{\texttt{full}}\}$. Figure 10 shows the loss statistics where the indexes are sorted by the loss value. As shown in the figure, the distribution of the sample importance is quite similar to the Pareto distribution (or the long tail distribution; as in the main text), indicating that selecting the elements with high scores can represent the full context set.

## D.5 Comparison with Memory-Efficient Meta-Learning Methods

Table 8: Comparison with first-order gradient-based meta-learning schemes on SIREN meta-learned under the CelebA ($178 \times 178$) dataset.

| Method | $K_{\text{train}}$ | $K_{\text{test}}$ | PSNR |
|---|---|---|---|
| FOMAML [15] | 16 | 16 | 25.43 |
| Reptile [41] | 16 | 16 | 32.98 |
| **GradNCP (Ours)** | 16 | 16 | **39.80** |
| FOMAML [15] | 128 | 128 | 37.27 |
| Reptile [41] | 128 | 128 | 39.87 |
| **GradNCP (Ours)** | 16 | 128 | **46.50** |

We also compare GradNCP with other efficient optimization-based meta-learning schemes, including first-order MAML (FOMAML) [15] and Reptile [41]. As shown in Table 8, GradNCP significantly outperforms the baselines by a large margin, even when we consider extremely long-horizon training for first-order meta-learning schemes, i.e., we consider first-order methods with 8 times longer adaptation during meta-training than GradNCP. This observation is consistent with prior works [11], which have also shown that first-order meta-learning schemes tend to struggle with NF learning tasks. Given this, we believe that efficient second-order meta-learning techniques such as GradNCP will be a promising direction in this field.

## D.6 Context Set Selection for Bootstrapping

Table 9: Comparison of context set choice for generating the target model on SIREN meta-learned with CelebA (178×178). We consider no target correction (None), random sampling (Random), gradient norm-based pruning (Norm-based), and the full context (Full) when adapting the target.

| Context set | PSNR (↑) | SSIM (↑) | LPIPS (↓) |
|---|---|---|---|
| None | 37.49 | 0.952 | 0.017 |
| Random | 37.56 | 0.951 | 0.016 |
| Norm-based | 38.19 | 0.958 | 0.013 |
| Full (Ours) | **38.90** | **0.967** | **0.009** |

We performed an experiment to investigate whether the gain of the bootstrapped correction mainly comes from the information recovery (by using the full context set) or the longer horizon effect. To this end, we examined various selection schemes for adapting the full context target, including random context (online), gradient norm-based pruned context, and full context set. The results, presented in Table 9, indicate that adaptation using the full context set is indeed effective, and the improvement is attributed to the recovery of information by the target model. Note that even random sampling and gradient norm-based pruning also improve the performance as they additionally provide more samples but less so than the full context set.

## D.7 Cross-domain Adaptation Experiments

Table 10: Cross-domain reconstruction performance (PSNR; dB) of meta-learned SIREN under UCF-101 (128×128×16) dataset. We adapt the network to a different dataset and modalities.

| Modality | Dataset | Method | PSNR (↑) |
|---|---|---|---|
| Image | CelebA (128×128) | Learnit / MAML [61] | 27.74 |
| | | **GradNCP (Ours)** | **28.57** |
| | Imagenette (128×128) | Learnit / MAML [61] | 25.18 |
| | | **GradNCP (Ours)** | **26.32** |
| Video | Kinetics-400 (128×128×16) | Learnit / MAML [61] | 26.42 |
| | | **GradNCP (Ours)** | **27.41** |

We also consider the cross-domain adaptation scenario: we adapt the meta-learned model on different datasets or even different modalities from the meta-training. In particular, we train our method on UCF-101 and adapt to two different image datasets (CelebA and Imagenette) and one video dataset (Kinetics-400) [28]. Table 10 summarizes the results: GradNCP significantly improves the performance over the baseline, indicating GradNCP has learned a transferable initialization from the diverse motion of UCF-101.

## D.8 Additional Comparison with TransINR under Test-time Adaptation

Table 11: Comparison with TransINR under same test-time adaptation (TTA) steps on SIREN meta-learned with CelebA (178×178) dataset. We further adapt the same adaptation steps (as GradNCP) from the predicted network from TransINR.

| Method | PSNR (↑) | SSIM (↑) | LPIPS (↓) |
|---|---|---|---|
| TransINR [6] | 32.37 | 0.913 | 0.068 |
| TransINR [6] + TTA | 34.12 | 0.932 | 0.046 |
| **GradNCP (Ours)** | **40.60** | **0.976** | **0.005** |

To further verify the superiority of GradNCP, we additionally compare with the test-time adaptation performance of TransINR. Here, we use SGD with the same adaptation steps (as GradNCP) to further optimize the predicted NF by TransINR. As shown in Table 11, GradNCP significantly shows better results even under this scenario. Still, note that such a comparison is not fair for GradNCP (in terms of # of parameters), as TransINR additionally uses a large Transformer encoder over GradNCP.

## D.9 Effectiveness of Using Full Context Set for Meta-testing

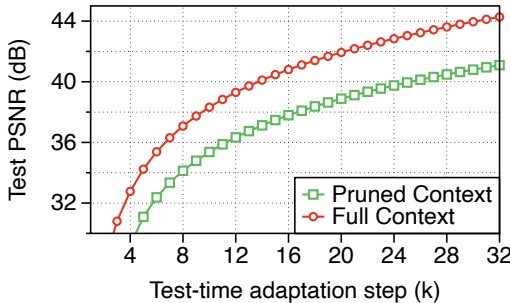

Figure 11: Comparison of test reconstruction performance between the utilization of full context set (Full Context) and gradient norm-based pruned context set (Pruned Context) during meta-test. The experiment is conducted over a meta-learn SIREN on CelebA (178×178) dataset with GradNCP.

To verify that the utilization of a full context set for meta-testing is truly effective, we compare the adaptation performance when using full and the gradient norm-based pruned context set. Here, we only apply the gradient scaling when using the full context set, as pruned context set gradient norm mismatch does not occur between the meta-training and testing. As shown in Figure 11, using the full context set for the meta-test time is indeed effective and shows consistent improvement over the pruned context set. Note that using pruned context set is also quite effective as it shows better adaptation performance than Learnit on a long adaptation horizon.

## D.10 GradNCP for General Meta-Learning Setup

Table 12: Many-shot in-domain adaptation accuracy (%) on mini-ImageNet datasets. We train Conv4 on 5-way and 10-way classification by using the same memory constraint and use the same number of inner-adaptation steps during meta-testing for a fair comparison. Reported results are averaged over three trials, subscripts denote the standard deviation, and bold denotes the best result.

| Method | 5-way 100-shot | 10-way 50-shot |
|---|---|---|
| MAML [15] | $66.03_{\pm 0.82}$ | $48.95_{\pm 0.52}$ |
| **GradNCP (Ours)** | $\mathbf{73.45_{\pm 0.83}}$ | $\mathbf{55.71_{\pm 0.49}}$ |

While we primarily focused on resolving the memory issue of meta-learning NFs due to their unique aspects different from common meta-learning setups (i.e., training with extremely large context sets which suffers from severe memory burden), we show that GradNCP can be used for general meta-learning setups that uses second-order gradient-based meta-learning. To this end, we consider a many-shot classification (e.g.,100-shot) scenario on mini-ImageNet to verify the efficacy of GradNCP. As shown in Table 12, GradNCP significantly outperforms MAML under the given memory constraint, e.g., 66.03 → 73.45 %. Based on this, we believe exploring GradNCP for general meta-learning setups also has great potential for application that requires second-order gradient-based meta-learning, e.g., learning automatic reweighting [46, 24].

## D.11 More Qualitative Comparison with Baselines on High-resolution Signals

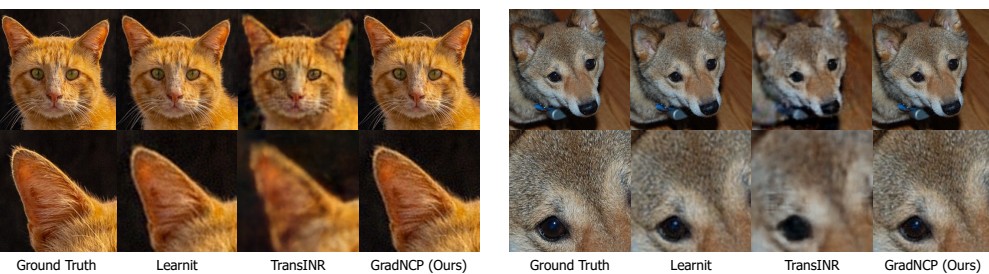

Figure 12: Qualitative comparison between GradNCP and baselines on AFHQ (512×512).

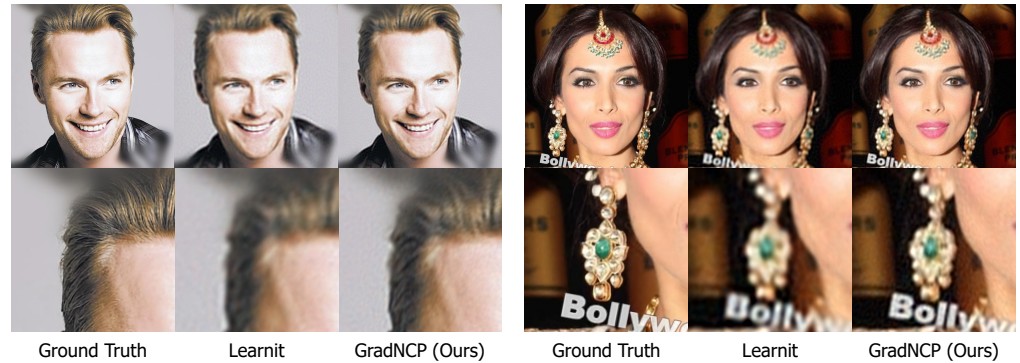

| Ground Truth | Learnit | GradNCP (Ours) | Ground Truth | Learnit | GradNCP (Ours) |

Figure 13: Qualitative comparison between GradNCP and baselines on CelebA-HQ (1024×1024). We compare GradNCP with Learnit as TransINR suffers from the out-of-memory issue.

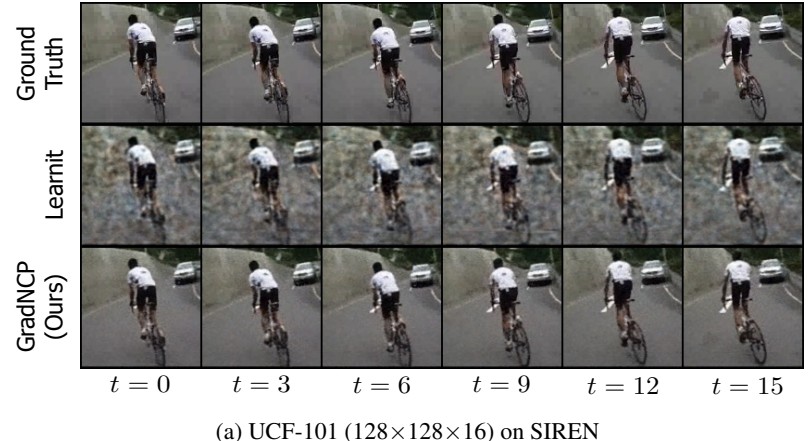

(a) UCF-101 (128×128×16) on SIREN

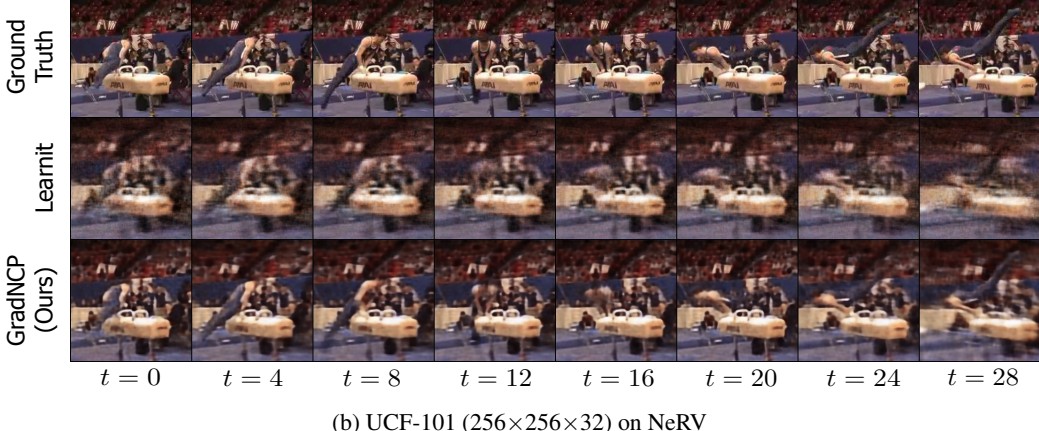

(b) UCF-101 (256×256×32) on NeRV

Figure 14: Qualitative comparison between GradNCP and Learnit on UCF-101 dataset.

# E More Visualizations of Selected Context Points via GradNCP

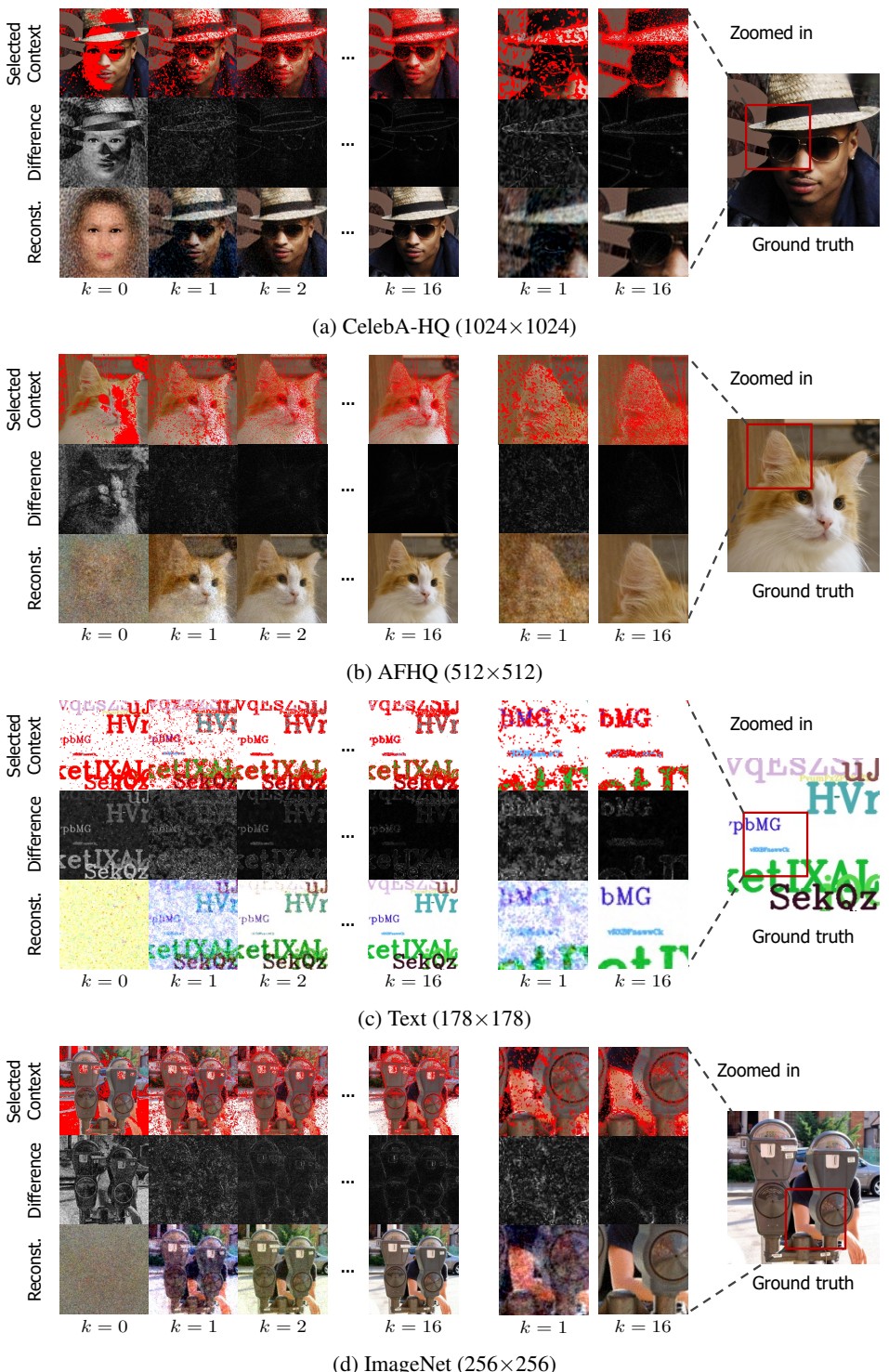

Figure 15: Visualization of selected context points (top), residuals relative to the original signal (middle), and the reconstruction (bottom) of GradNCP trained on (a) CelebA-HQ, (b) AFHQ, (c) Text, and (4) ImageNet. Selected coordinates are highlighted in red (top) where our selection scheme focuses on only 25% of available points at each of the $k$ adaptation steps. GradNCP first focuses on modeling global structure while subsequently learning high-frequency details.

