# OpenReview forum: "Learning Large-scale Neural Fields via Context Pruned Meta-Learning"
_NeurIPS.cc/2023/Conference — NeurIPS 2023 poster_

### Official Review · Reviewer_9F3K · 2023-07-04

**Soundness:** 3 good
**Presentation:** 3 good
**Contribution:** 2 fair
**Rating:** 5
**Confidence:** 3

**Summary:**

This paper proposes a way to sample the subset of the dataset to improve convergence speed and the eventual accuracy in the meta-learning framework, especially for neural fields. They also propose a bootstrap correction to resolve the well-known myopia issue. Moreover, gradient re-scaling during meta-testing also improved the performance. They have provided the experimental results on nine datasets, showing superior performance compared to the baseline methods.

**Strengths:**

1. This paper is well-written and easy to follow. The method is simple and easy to apply in the existing meta-learning frameworks, and the performance improvements are not marginal.

2. It also provides very comprehensive experimental results, showing the effectiveness of each component very well (although I have some concerns, see the weakness section).


**Weaknesses:**

Here, I am writing both weaknesses and the questions in this section since both are quite related (but not necessarily all questions are weaknesses). Depending on the rebuttal and other reviewers' opinions, I would revise my score.

1. Although the authors have provided extensive experimental results, I think the current setup would not clearly reveal the actual effects of the newly proposed components. First, the main contribution of this paper is ‘norm-based context pruning’, and I believe this paper should have compared against other data pruning or importance sampling methods (for example, you can refer to [1] for more related works on this). At least, the authors could have compared gradNCP to simple loss-based top-k sampling, which is much simpler than the proposed method. They are using the norm of the gradient of the last layer to measure the importance, and I think the higher loss, the higher the gradient norm. Hence, simple loss-based top-k sampling would also work well.

2. I strongly encourage the authors to provide enough related works on data pruning or the importance sampling studies.

3. The effect of ‘meta-training w/ gradNCP’ was not adequately tested. The proper setup would be (Random initialization + test-time gradNCP) vs. (MAML + test-time gradNCP) vs. the proposed method. In this way, the authors can convince that meta-training w/ gradNCP is really necessary and effective.

4. In L145-146: I respectively disagree with this statement. The authors have only tested ‘reconstruction’ tasks, arguing that generalization performance is not important in neural fields. I think the most interesting applications of neural fields are related to generalization, such as novel view synthesis (NeRF) or 3d surface reconstructions (not just point-wise reconstructions). And IMHO, the current image and video reconstruction experiments done in this paper are very far from practical usage. The compression rate is nowhere close to the state-of-the-art image and video codecs, and the encoding time for a higher-compression rate is too slow to use in practice.

5. Why only neural fields setup? The main claim of the paper is the effectiveness of the gradNCP in neural fields, but I do not see any specific treatment for neural fields. In other words, gradNCP can apply to any meta-learning setup? This question is not necessarily a criticism but more on the possibility of strengthening the paper.

[1] Not All Samples Are Created Equal: Deep Learning with Importance Sampling, Katharopoulos et al., ICML 2018.

-minor comments-
L128: C^(i) was the set of coordinates of i-th signal, but here C^k denotes the set of coordinates of k-th step.
L142: {x,y} -> ({x,y})


**Questions:**

The questions are embedded in the weakness section.

**Limitations:**

I do not see any potential negative societal impact from this manuscript.

---

> ### Author Rebuttal · Authors · 2023-08-07
>
> Dear reviewer 9F3K,
>
> We sincerely appreciate your efforts and insightful comments to improve our manuscript. We respond to each of your comments one-by-one in what follows.
>
> ---
>
> **[W1] Comparison with other data pruning. e.g., loss-based top-K selection.**
>
> First, we carefully remind the reviewer that we already compared with other data pruning methods such as SPG and TPG [1] (see Figure 3 (c)), where GradNCP shows almost the same performance while significantly reducing the time complexity for data selection.
>
> Moreover, we are very happy that the reviewer has pointed out the loss-based selection. During the initial development, we also tried the loss-based selection, showing a good performance as well, where GradNCP consistently exhibited better performance (see Table 4 in the attached pdf, e.g., 33.99$\to$35.28 dB in UCF-101). We believe this is because the loss-based selection is an approximation of GradNCP and thus GradNCP can provide better data pruning results. Specifically, assuming that the penultimate feature has a bounded norm, i.e., $||[\phi(.)~ 1]|| \le M$ where $M\in R^{+}$, the GradNCP score can be approximated to loss-based selection as follows: $||(y-f(x)) [\phi(x)~ 1]^{T} || \le ||y-f(x)|| ~ || [\phi (x)~1] || \le M ||y-f(x)||$. Since some setups consider SIREN (i.e., sine activation), the assumption holds for these setups, thereby showing similar performance in some cases, e.g., CelebA. We thank the comment and will describe the relationship between GradNCP and the loss-based selection in the revision.
>
> ---
>
> **[W2] More related works on data pruning.**
>
> Thank you for the comment. In addition to our related work section on data pruning in Appendix C, we will revise our manuscript with more comprehensive studies of data pruning literature, including the reference that you mentioned [2].
>
> ---
>
> **[W3] Effect of meta-training with GradNCP is not tested due to test-time GradNCP: Use test-time GradNCP for MAML.**
>
> We would like to clarify that the only difference between MAML and GradNCP for meta-testing is the gradient re-scaling which *has no effect for MAML*. Specifically, gradient re-scaling is proposed to match the gradient step size difference of meta-training and meta-testing (as GradNCP use pruned set for meta-training and a full set for meta-testing), while MAML uses the full set in both stages. Hence, the re-scaling does not affect any performance of MAML as the re-scaling constant is equal to 1.
>
> ---
>
> **[W4] Clarification of reconstruction experiments and GradNCP for NeRFs.**
>
> Fast reconstruction of NFs is an important basis for multiple applications in NF research. For example, a recently emerging paradigm in exploiting NFs, first converts signals into NFs by reconstructing them and casting them for various downstream tasks (e.g., generation or compression [3,4]). In particular, these approaches have recently shown great promise, e.g., [4] outperforms image, video, and audio codecs in compression, [5] achieves state-of-the-art results in 3D shape generation by reconstructing (not generalizing) each of 3D shape data and do generative modeling in this NF weight space, to name a few. Hence, prior works highlight the importance of the *fast* reconstruction of NFs [3,4,5], where we also followed the prior work’s experimental setup [6].
>
> Nonetheless, we also agree with the reviewer that generalization is also an important application of NFs. To this end, we apply GradNCP for the NeRF experiment on ShapeNet Car to verify the generalization performance of GradNCP. Here, following the setup in [6], we train NeRFs with a given single view of the 3D scene and evaluate the performance on other unseen views. As shown in Table 2 in the attached pdf, GradNCP significantly outperforms other baselines including Learnit (e.g., 22.80 $\to$ 24.06 dB), and even achieves better performance than TransINR which utilizes an additional (large) Transformer. We believe using GradNCP for large-scale generalization of NFs (e.g., NeRF for Google street view) will be an interesting future direction to explore, where we believe the efficiency of GradNCP can contribute to this field.
>
> ---
>
> **[W5] GradNCP for general meta-learning setup.**
>
> Thank you for your insightful comment! We primarily focused on resolving the memory issue of meta-learning NFs due to their unique aspects different from common meta-learning setups, i.e., training with extremely large context sets (can be >1 million) which suffers from severe memory burden. Note that this issue easily becomes more fatal since second-order gradients are necessary for meta-learning NFs [3] in contrast to other meta-learning problems that sometimes show comparable performance with memory-efficient first-order methods [7].
>
> Nonetheless, we also agree that GradNCP can be used for general meta-learning setups. Following your suggestion, we consider a many-shot classification (e.g.,100shot) scenario on mini-ImageNet to verify the efficacy of GradNCP. As shown in Table 1 in the attached pdf, GradNCP significantly outperforms MAML under the given memory constraint, e.g., 66.03$\to$73.45 \%. It is worth noting that this also highlights the *generalization ability* of GradNCP and we believe exploring GradNCP for general meta-learning setups has great potential.
>
> ---
>
> **Editorial comments.**
>
> Thank you very much and we will update typos in the revision.
>
> ---
>
> **Reference**\
> [1] Automated Curriculum Learning for Neural Networks, ICML 2017\
> [2] Not All Samples Are Created Equal: Deep Learning with Importance Sampling, ICML 2018\
> [3] From Data to Functa: Your Data Point is a Function and You Can Treat it Like One, ICML 2022\
> [4] Modality-Agnostic Variational Compression of Implicit Neural Representations, ICML 2023\
> [5] HyperDiffusion: Generating Implicit Neural Fields with Weight-Space Diffusion, ICCV 2023\
> [6] Learned Initializations for Optimizing Coordinate-Based Neural Representations, CVPR 2021\
> [7] On First-Order Meta-Learning Algorithms, arXiv 2018

---

> > ### Comment · Reviewer_9F3K · 2023-08-14
> >
> > I appreciate your efforts in answering my questions. Here are two more follow-up questions.
> >
> > **[W1] GradNCP vs. Loss-based data selection:** You provided the results of 4 datasets. I do appreciate it, but how about others? are the performance improvement is consistent?
> >
> > **[W3] Is meta-learning necessary?:** I was curious if 'meta-learning' is necessary. From my understanding, this paper has two technical contributions, meta-learning and context-pruning. I see context-pruning works. How about meta-learning? GradNCP vs. (Random init + context-pruning using your methods, no meta-learning) would be a good ablation study.

---

> > > ### Author Response · Authors · 2023-08-14
> > > **Thank you for the response**
> > >
> > > Dear reviewer 9F3K,
> > >
> > > We sincerely thank the reviewer for the prompt response and efforts in reading our response. We would like to respond to the remaining concerns in what follows.
> > >
> > > ---
> > >
> > > **[W1] More datasets when comparing loss-based and gradient norm-based selection.**
> > >
> > > Following the reviewer’s request, we are currently running additional experiments on more datasets to compare loss-based and gradient norm-based selection. We will share the additional results as soon as the training is done.
> > >
> > > ---
> > >
> > > **[W3] Necessity of meta-learning: comparison with random initialization.**
> > >
> > > First, we want to clarify that we don’t need to consider context pruning in the case of random initialization. Specifically, our context pruning is designed for **only** a memory-inefficient “meta-training” stage (i.e., “meta-learn” a better initialization) in a memory-efficient manner; thus it is not related to random initialization that does not have any meta-learning. Moreover, note that during test-time, memory usage is identical for all baselines (irrespective of whether the baseline is meta-trained or not) and it is not intensive as well; thus we used the full context set (without pruning) for evaluating all methods. Consequently, the result in Table 1 in our manuscript already verifies the necessity of meta-learning, as in this table, we showed all meta-learning schemes always significantly outperform random initialization using exactly the same test scheme (i.e., with the full context set).
> > >
> > > Nevertheless, following your suggestion, we also compare the performance between GradNCP vs. [random init. + context pruning]. As shown in the table below, GradNCP significantly outperforms [random init. + context pruning] in all cases on reconstruction quality measured with PSNR (dB), which further highlights the importance of “meta-learning” in our method.
> > >
> > > \begin{array}{llcccc}
> > > \hline
> > > \text{Initialization}  & \text{Context set used for test-time} & \text{CelebA}  & \text{Imagenette} & \text{AFHQ} & \text{ImageNet} \newline
> > > \hline
> > > \text{Random} & \text{Pruned context} & 18.42 & 18.31 & 17.88 & 17.83 \newline
> > > \text{Random} & \text{Full context} & 19.94 & 18.57 & 18.57 & 18.72 \newline
> > > \text{GradNCP} & \text{Full context} & \mathbf{40.60} & \mathbf{38.72} & \mathbf{29.61} & \mathbf{32.52} \newline
> > > \hline
> > > \end{array}

---

> > > > ### Author Response · Authors · 2023-08-17
> > > > **Thank you for the response**
> > > >
> > > > Dear reviewer 9F3K,
> > > >
> > > > We sincerely thank again the reviewer for your prompt response and efforts in reading our response. We would like to respond to the remaining concerns in what follows.
> > > >
> > > > ---
> > > >
> > > > **[W1] More datasets when comparing loss-based and gradient norm-based selection.**
> > > >
> > > > Following the reviewer’s suggestion, we provide the comparison between gradient-based and loss-based selection on all datasets (except for ImageNet which takes more than 10 days to train). As shown in the table below, our gradient-based selection mostly outperforms the loss-based selection (13 out of 16 cases) where our method shows significant improvements in some cases, e.g., our improvement of 1.5 dB in UCF-101 (128) on NeRV is quite significant as PSNR is a log-scale metric (i.e., reduced the reconstruction loss more than 30 times). Also, note that gradient-based selection shows at least comparable performance for the other three cases as well.
> > > >
> > > > Here, we emphasize that our gradient-based selection can provide more accurate results than the loss-based selection, regardless of neural field (NF) architectures. This is because the loss-based selection is an approximation of gradient-based selection where this approximation error is small if the norm of the penultimate feature is bounded—however, such a condition holds for a few cases (e.g., SIREN that uses sine activations) and does not always hold for general NF architectures that use ReLU activations (e.g., NeRV and FFN). This can be empirically validated with the results we provided: gradient-based selection shows a **consistent improvement** than loss-based selection under NF architectures with ReLU activations. Thanks for the incisive comments and we will add this discussion in the final draft.
> > > >
> > > >
> > > > \begin{array}{l | llllllllll}
> > > > \hline
> > > > \text{Architecture}  & \text{SIREN} \newline
> > > > \text{Dataset}  & \text{CelebA} &\text{Imagenette} &\text{Text} & \text{AFHQ} &\text{CelebA-HQ} &\text{UCF-101 (128)} &\text{UCF-101 (256)} &\text{Libri (1sec)} &\text{Libri (3sec)} &\text{ERA5}  \newline
> > > > \hline
> > > > \text{Loss} & 40.54 & 37.71 & \mathbf{33.11} & 29.37 & 28.89 & 26.59 & 22.76 & \mathbf{43.40} & \mathbf{36.45} & 74.10 \newline
> > > > \text{Grad} & \mathbf{40.60} & \mathbf{38.72} & 32.33 & \mathbf{29.61} & \mathbf{28.90} & \mathbf{26.92} & \mathbf{22.92} & 43.25 & 36.24 & \mathbf{75.11} \newline
> > > > \hline
> > > > \end{array}
> > > >
> > > > \begin{array}{l |ll | llll}
> > > > \hline
> > > > \text{Architecture}  & \text{NeRV} & & \text{FFN} &  &  &   \newline
> > > > \text{Dataset}  & \text{UCF-101 (128)} &\text{UCF-101 (256)}  & \text{CelebA} &   \text{Text}  &   \text{Libri (1sec)} &  \text{Libri (3sec)}   \newline
> > > > \hline
> > > > \text{Loss} & 33.99 & 28.58 & 28.81 & 21.93 & 33.78 & 31.56 \newline
> > > > \text{Grad} & \mathbf{35.28} & \mathbf{28.65} & \mathbf{29.36} & \mathbf{22.52} & \mathbf{33.93} & \mathbf{32.04} \newline
> > > > \hline
> > > > \end{array}
> > > >
> > > > ---
> > > >
> > > > **[W3-2] This paper has two technical contributions.**
> > > >
> > > > We further would like to remark that GradNCP consists of multiple (technical) contributions in addition to online context pruning for meta-learning; (i) bootstrapped correction and (ii) gradient re-scaling are also our key contributions, where we validate the importance of each component in Section 3.2 and Appendix D.2, respectively.

---

> > > > > ### Comment · Reviewer_9F3K · 2023-08-18
> > > > >
> > > > > Thanks for your efforts to answer the questions, I do appreciate it.
> > > > >
> > > > > Given the results you provided, I do not think there is a meaningful difference between Loss vs Grad (except three datasets, Imagenette (Grad wins), Text (Loss wins), and ERA5 (Grad winds). it seems that Grad works well w/ FFN architecture, but we can just use SIREN who performs much better?). Since hyperparameters affect performance a lot (we all know about this), I do not think those improvements are sufficient to show the effectiveness of the proposed "grad-based" method.
> > > > >
> > > > > It seems that gradNCP works well on UCF-101 (128). But in UCF-101 (256), it shows only 0.07 difference. Note that the title is 'Learning large-scale Neural Fields...', and I think UCF-101 (256) is a more representative dataset in this context.
> > > > >
> > > > > I believe this paper has contributions, but I am worried this paper may mislead the readers. The name of the method is 'gradNCP', but a much simpler (and probably more scalable) loss-based pruning performs about the same. Assuming that the authors will revise the main text, **not highlighting 'grad-based' methods**, I raised my rating, 4->5.

---

> > > > > > ### Author Response · Authors · 2023-08-18
> > > > > > **Thank you very much for your response**
> > > > > >
> > > > > > Dear reviewer 9F3K,
> > > > > >
> > > > > > We sincerely thank again the reviewer for your prompt response and efforts in reading our response.
> > > > > >
> > > > > > Also, we thank the overall positive review of our paper.\
> > > > > > Due to your valuable and constructive suggestions, we do believe that our paper is much improved.
> > > > > >
> > > > > > Following your suggestion, we will highlight the effectiveness of loss-based selection and the connection with our gradient-based selection in the revised manuscript. For your interest, we actually used the loss-based selection at an early stage of developing our method. We later decided to use gradient-based selection (instead of the loss-based selection) as (i) it is more (often significant) effective overall (not always though) regardless of NF model choices (as there are several recent NF architectures use ReLU [1,2]) and (ii) also only requires a single forward pass to compute. Here, we emphasize that we use the same hyper-parameters for both loss-based and gradient-based selection, and the superiority of the latter is not from hyper-parameter tuning (i.e., hyper-parameters are not specially tuned for gradient-based; we rather used the ones found during initial development with loss-based selection)
> > > > > >
> > > > > > Finally, we remind that our major contribution is to propose the overall meta-learning pipeline for large-scale NF training, i.e., scaling meta-learning NFs through data pruning and introducing a correction method that fixes possible information loss made by the data pruning. Developing a better design choice in our framework, e.g.,  data pruning scheme, than what we used in this paper is definitely an interesting future direction to explore.
> > > > > >
> > > > > > We sincerely thank the reviewer again for reviewing our paper, taking the time to read the response, and giving great suggestions to improve our manuscript.
> > > > > >
> > > > > > Thank you very much,\
> > > > > > Authors
> > > > > >
> > > > > > **Reference**\
> > > > > > [1] Instant Neural Graphics Primitives with a Multiresolution Hash Encoding, SIGGRAPH 2022\
> > > > > > [2] ReLU Fields: The Little Non-linearity That Could, SIGGRAPH 2022

---

### Official Review · Reviewer_3CuA · 2023-07-05

**Soundness:** 3 good
**Presentation:** 2 fair
**Contribution:** 3 good
**Rating:** 7
**Confidence:** 3

**Summary:**

This paper presents an efficient optimization-based meta-learning technique for large-scale neural field training. It introduces automated online context point selection, resulting in significant memory savings and improved model quality. The authors also propose a bootstrap correction to enhance the meta-learned initialization, reducing errors introduced by reduced context sets and addressing the myopia of optimization-based meta-learning. The method achieves state-of-the-art results in signal reconstruction across multiple modalities, providing insights into the algorithmic components of the approach.

**Strengths:**

The paper demonstrates several strengths across the dimensions of originality, quality, clarity, and significance.

**Originality**: This paper introduces Gradient Norm-based Context Pruning (GradNCP). This original approach combines curriculum learning and data pruning techniques to address the computational burden and myopia issues in optimization-based meta-learning for neural field training.

**Quality**: The paper demonstrates high technical quality through a well-formulated GradNCP framework and extensive empirical evaluation, showcasing state-of-the-art results on multiple datasets and modalities. The algorithmic component analysis further enhances the proposed method's quality and understanding.

**Clarity**: The paper effectively communicates its motivations, methods, and findings, providing a clear overview of neural fields and their challenges.

**Significance**: The paper is important in the field as it addresses memory limitations and myopia issues, providing a scalable and efficient solution for large-scale neural field training.

**Weaknesses:**


(1) **Clarity of motivation**:  The motivation behind choosing optimization-based meta-learning is not very clear.  Can the authors provide a more detailed explanation highlighting the advantages of optimization-based meta-learning for large-scale neural field training and its prior successful applications?

(2) **Lack of comparative experiments**: The paper does not include comparative experiments with other optimization-based meta-learning methods, such as MAML. To enhance comparability, the authors should consider conducting comparative experiments to evaluate GradNCP's performance relative to these methods in terms of efficiency, memory utilization, and reconstruction performance.

**Questions:**

(1) It would be helpful to understand the criteria or scoring mechanism used for online context point selection in GradNCP. How does the algorithm determine the importance of each data point and decide which subset to focus on during each learning iteration?

(2) The paper mentions the bootstrap correction to minimize errors introduced by reduced context sets. Could the authors provide more details on how the bootstrap correction works and how it improves the quality of the meta-learned initialization?

(3) Are there any specific limitations or trade-offs of GradNCP that should be considered? Are there scenarios or types of signals where GradNCP might not perform as effectively?


**Limitations:**

Yes

---

> ### Author Rebuttal · Authors · 2023-08-07
>
> Dear reviewer 3CuA,
>
> We sincerely appreciate your efforts and insightful comments to improve the manuscript. We respond to each of your comments one-by-one in what follows.
>
> ---
>
> **[W1] Clarification on the motivation for using optimization-based meta-learning.**
>
> The motivation for using optimization-based meta-learning is mainly two-fold; note that there mainly exist two meta-learning approaches for neural fields (NFs) training which are optimization-based and amortization-based meta-learning.
>
> First, optimization-based meta-learning is modality and model agnostic: it does not require any (domain-specific) encoder architectures, showing a versatile range of applications. On the other hand, amortization-based meta-learning (e.g., TransINR) often requires significant modification (or even limited) when using it for each modality and NF architecture. For instance, TransINR requires (a) tokenization, which is not straightforward to apply for spherical coordinate datasets (e.g., ERA5), and (b) notable modifications to the architecture when it consists of non-MLP layers (e.g., NeRV) as the framework is specific to MLPs (see Line 303).
>
> Moreover, optimization-based meta-learning is specialized for NF reconstruction as it directly optimizes the loss of the given context set. Meanwhile, amortization-based meta-learning is known to suffer from the context set underfitting [1,2] as it is difficult to predict the parameter that minimizes the loss at once, somewhat limiting the performance of the reconstruction. It is worth noting that MAML outperforms TransINR in most cases (see Table 1 in the main text).
>
> Finally, we remark prior works in meta-learning NFs, largely focus on using optimization-based meta-learning schemes due to these strengths [3,4,5]. Also, an emerging paradigm in NF research is to convert signals into NFs and then utilize them for downstream tasks (e.g., generation, compression, and classification [4,5,6]) where these works also leverage optimization-based meta-learning to train large-scale NFs.
>
> ---
>
> **[W2] Comparison with other optimization-based meta-learning methods such as MAML.**
>
> We carefully remark that we already extensively compared GradNCP with the optimization-based meta-learning method, i.e., MAML (note that Learnit and MAML are equivalent), under multiple evaluation metrics including, memory efficiency (see Figure 3 (a)), run-time efficiency (see Figure 3 (b)) and reconstruction performance (all tables in the main manuscript). Furthermore, we additionally compared with other optimization-based meta-learning schemes in Appendix D.4 (e.g., first-order MAML and Reptile); we will explicitly mention it in our final manuscript.
>
> ---
>
> **[Q1] Question regarding the selection criteria.**
>
> We focus each learning step on the subset of data with the highest expected immediate improvement in model performance, which is approximated efficiently as the gradient norm w.r.t the last layer (see the Equation after 154 in the main paper), i.e., we select the samples that have the higher gradient norm. We show that such criteria only require a single forward pass to compute (see Eq. (2)), thereby can be utilized in an *online* manner (i.e., iteratively use during training) due to its computational efficiency.
>
> ---
>
> **[Q2] How Bootstrapped correction improves the performance.**
>
> Bootstrapped correction improves the performance as it regularizes the parameter adapted with the pruned context $\theta_K$ to be close to a better-performing parameter (which is the bootstrapped target $\theta_{K+L}^{\text{boot}}$). Note that bootstrapped target is further optimized $L$ steps with *the full context set*, hence, it is not only a better-performing parameter but also contains the information of pruned-out context points. We experimentally verify that the performance of bootstrapped target $\theta_{K+L}^{\text{boot}}$ is consistently better than $\theta_K$ in Figure 5 (c), providing a well-conditioned target for the regularization (also see the effect of using the full context set in Appendix D.6).
>
> ---
>
> **[Q3] Limitations or trade-offs of GradNCP.**
>
> While GradNCP is possible to meta-learn on the highest resolution signal than the prior works, still it might be challenging to meta-learn NFs on exceptionally high-resolution signals (e.g., long 8K movie) where even a *single* forward pass is even not possible. We believe that a great variety of techniques can be developed in this direction, e.g., iterative searching of coordinates with high gradient norms by starting from a low-dimension grid and incrementally increasing the resolution of the sampled area.
>
> ---
>
> **Reference**\
> [1] Attentive Neural Processes, ICLR 2019\
> [2] Neural Processes with Stochastic Attention: Paying More Attention to the Context Dataset, ICLR 2021\
> [3] Learned Initializations for Optimizing Coordinate-Based Neural Representations, CVPR 2021\
> [4] COIN++: Neural Compression Across Modalities, TMLR 2022\
> [5] Your data point is a function and you can treat it like one, ICML 2022\
> [6] Modality-Agnostic Variational Compression of Implicit Neural Representations, ICML 2023

---

> > ### Comment · Reviewer_3CuA · 2023-08-16
> >
> > I am highly satisfied with the authors' responses, as they have thoroughly addressed my concerns. Taking into account the feedback from other reviewers, I recommend accepting this paper.

---

> > > ### Author Response · Authors · 2023-08-16
> > > **Thank you for the response**
> > >
> > > Dear reviewer 3CuA,
> > >
> > > Thank you for letting us know! We are delighted to hear that our rebuttal addressed your questions well.\
> > > Also, thank you for reading other reviewers' comments and feedback as well.
> > >
> > > If you have any further questions or suggestions, please do not hesitate to let us know.
> > >
> > > Thank you very much,\
> > > Authors

---

### Official Review · Reviewer_EMYi · 2023-07-06

**Soundness:** 3 good
**Presentation:** 3 good
**Contribution:** 3 good
**Rating:** 6
**Confidence:** 3

**Summary:**

This paper addresses the issues of high memory usage and myopia problem in optimization-based meta-learning for large-scale neural field training. Three techniques are proposed: 1) online context pruning improves memory efficiency by using a selected context set in the inner learning problem, 2) bootstrap correction counteracts the information loss introduced by the context pruning, and 3) gradient scaling is introduced at meta-test time to allow the use of full context set. Experiments have demonstrated improved memory efficiency and model performance of the proposed method in comparison with existing models.

**Strengths:**

The presentation of the work is clear and well-structured: The problem setting of memory efficiency in optimization-based meta-learning on large-scale tasks is important; The techniques of context pruning and bootstrap correction are straightforward and sound; Empirical results are also extensive.

**Weaknesses:**

1. It looks like the derivation of Equation 2 relies on the linearity of the last output layer. How about the setting when the final layer is a more general function? Is Equation 2 applicable in such a general setting?
2. Could the author explain the jittering in Figure 3c and 5c? From the two figures, it seems that the training of the meta-model is not stable.
3. I think the ablation study could be more sufficient. In Figure 5b, does the “test-time adaptation steps” refer to $K_{test}$ or $L$? There should be ablation studies on $L$ in bootstrap correction and the effect of gradient rescaling at test time.

**Questions:**

Please check the weaknesses mentioned above.

**Limitations:**

Yes

---

> ### Author Rebuttal · Authors · 2023-08-07
>
> Dear reviewer EMYi,
>
> We sincerely appreciate your efforts and insightful comments to improve the manuscript. We respond to each of your comments one-by-one in what follows.
>
> ---
>
> **[W1] GradNCP for non-linear last layer architectures.**
>
> We follow the setup of common design choice for NFs [1,2,3,4] that uses the linear output layer and derive Eq. 2. For the non-linear last layer, one can simply calculate the gradient norm w.r.t last as well as follows:
>
> $\Big\lVert \big(y- f (x) \big) \sigma'(W \phi(x) + b)\big[\phi(x), \mathbf{1} \big]^{\mathsf{T}}  \Big\rVert$\
> where the output of NF is $f(x) = \sigma(W \phi (x) + b)$, $\sigma$ is the non-linear function, e.g., Sigmoid, and $\sigma'(x)=\frac{d\sigma(x)}{dx}$. Note that this score also requires only a *single* forward pass to compute, thereby highly efficient. We thank the comment and in the revision, we will mention that Equ. 2 is for NFs with the linear output layer and add the score function for the non-linear output layer as well.
>
> ---
>
> **[W2] Instability in the meta-training process.**
>
> We first note that such an unstable training process is quite prevalent in meta-learning NFs, e.g., in the conclusion section of [5] and Figure 12 of [6]. Specifically, it is a common observation in meta-learning that models without non-skip connections may suffer from training instability [7] (note that many NFs do not use skip connections [1,2]). We believe a similar phenomenon is taking place in our experiments. Still, we emphasize that the final performance is usually not affected by the instability during meta-training, and the performance gain of GradNCP is significant compared to the baselines while sharing the same instability issue.
>
> Nevertheless, we think stabilizing meta-learning NFs is an important and interesting future direction where GradNCP can also be beneficial in this direction. Given that the prior work highlighted the importance of task batch size for stabilizing meta-learning NFs (Appendix B.2 of [2]), we believe increasing the task batch size with the reduced memory by context pruning with GradNCP can be an effective approach for stable training.
>
> ---
>
> **[W3-1] Clarification of Figure 5b.**
>
> We clarify that the “test-time adaptation steps” refer to $K_{\text{test}}$ (not $L$): note that we do not use bootstrapped correction during meta-testing.
>
> ---
>
> **[W3-2] Ablation study of test-time gradient re-scaling.**
>
> Thank you very much for pointing it out. We have the ablation study of test-time gradient re-scaling in Appendix D.2; we will explicitly mention it in our final manuscript. In this experiment, we showed how the using test-time gradient re-scaling affects the reconstruction performance when using a full context set at the meta-test time. We also provide the analysis that the magnitude of the gradient norm significantly differs when using full context and pruned context sets, which indicates the necessity of test-time gradient re-scaling.
>
> ---
>
> **[W3-3] Ablation study of $L$ in bootstrapped correction.**
>
> Thank you very much for your constructive comment. Following the reviewer’s suggestion, we perform an ablation study regarding the step for generating bootstrapped target $L$. As shown in Table 3 in the attached pdf, we find that the performance rapidly improves until $L=5$ and observed moderate improvement by further increasing $L$. During the development, we found that $L=5$ is already quite effective across all modalities and datasets, and we believe tuning the hyperparameter will improve the performance.
>
> ---
>
> **Reference**\
> [1] Implicit Neural Representations with Periodic Activation Functions, NeurIPS 2020\
> [2] Fourier Features Let Networks Learn High Frequency Functions in Low Dimensional Domains, NeurIPS 2020\
> [3] NeRV: Neural Representations for Videos, NeurIPS 2021\
> [4] Scalable Neural Video Representations with Learnable Positional Features, NeurIPS 2022\
> [5] From data to functa: Your data point is a function and you can treat it like one, ICML 2022\
> [6] COIN++: Neural Compression Across Modalities, TMLR 2022\
> [7] How to train your MAML, ICLR 2019

---

> > ### Comment · Reviewer_EMYi · 2023-08-16
> >
> > Thank you for clarifying the details of the methodology and experiments and the additional ablation study. My concerns have been addressed.

---

> > > ### Author Response · Authors · 2023-08-16
> > > **Thank you for the response**
> > >
> > > Dear reviewer EMYi,
> > >
> > > We are very happy to hear that our rebuttal addressed your questions well.\
> > > Due to your valuable and constructive suggestions, we do believe that our paper is much improved.
> > >
> > > If you have any further questions or suggestions, please do not hesitate to let us know.
> > >
> > > Thank you very much,\
> > > Authors

---

### Official Review · Reviewer_hvG5 · 2023-07-07

**Soundness:** 3 good
**Presentation:** 3 good
**Contribution:** 3 good
**Rating:** 7
**Confidence:** 3

**Summary:**

The paper proposes a set of new techniques for optimizing neural fields with meta-learning. The new techniques includes using gradient norms to prune less important examples, bootstrapping to correct for myopia of the inner loop optimization, and gradient rescaling to correct for different context sizes at train and test time. The method shows improved reconstruction on a diverse set of data across multiple modalities. The paper also does ablation experiments and analysis to understand how each component affects performance.

**Strengths:**

- The paper is overall well written, and the method is straight-forward.

- Context-pruning for meta-learning seems like a generally applicable technique that could benefit applications beyond neural fields.

- GradNCP improves over current state-of-the-art neural field meta-learning techniques on most of the standard benchmarks.

- The ablation experiments and analysis give good insight on how each component of GradNCP affects reconstruction quality.


**Weaknesses:**

- I am unclear on the motivation given by the paper. The stated motivation for GradNCP is improved reconstruction quality using Neural Fields. The main application of neural fields and the one where reconstruction quality is the focus is neural radiance fields. However, the paper does not mention NeRF or do any evaluations on NeRF benchmarks. For reference, [1] runs experiments on the SRN Cars dataset. I’m wondering why this is the case, and if GradNCP cannot be used for NeRFs I think it should be made clear. If GradNCP improves reconstruction quality or optimization efficiency for NeRF this could be highly impactful.  If the motivation is learning higher quality reconstructions on high resolution images and videos, then my question is what is the motivation or application for that?

- I think the contributions for bootstrapped corrections are not made clear in the paper. In the introduction, the paper proposes bootstrap correction as one of the contributions, but does not mention [2]. Later the paper mentions [2] and say that it’s discussed in the related works, but I didn’t see any discussion of [2] in the related works in the appendix. It would help to clarify the papers contributions and discuss relevant work more thoroughly. For example, this paper [3] uses gradient-norms to select training examples.

1. *From data to functa: Your data point is a function and you can treat it like one*
2. *Bootstrapped Meta-Learning*
3. *Deep Learning on a Data Diet: Finding Important Examples Early in Training*



**Questions:**

- In meta-learning for classification literature, works have found that pretrained initializations with fine-tuning are competitive with meta-learning approaches. It would be useful to see the baseline of standard batch training without meta-learning in the context of NFs (i.e. use first-order gradients and no outer loop). Considering one motivation of GradNCP is to allow for larger context size, I would think standard training could perform well given that you can use as large of a context size as you want with gradient accumulation.



**Limitations:**

Authors adequately address the limitations of their method.

---

> ### Author Rebuttal · Authors · 2023-08-08
>
> Dear reviewer hvG5,
>
> We sincerely appreciate your efforts and comments to improve the manuscript. We respond to your comment in what follows.
>
> ---
>
> **[S2] GradNCP could be beneficial for meta-learning applications beyond NFs.**
>
> Thank you for your insightful comment! We primarily focused on resolving the memory issue of meta-learning NFs due to their unique aspects different from common meta-learning setups, i.e., training with extremely large context sets (can be >1 million) which suffers from extreme memory burden. Note that this issue easily becomes more fatal since second-order gradients are necessary for meta-learning NFs [1] in contrast to other meta-learning problems that sometimes show comparable performance with memory-efficient first-order methods.
>
> Nevertheless, we also agree that GradNCP can be used for general meta-learning setups. Accordingly, we consider a many-shot classification (e.g., 100-shot) scenario on mini-ImageNet to verify the efficacy of GradNCP. As shown in Table 1 in the attached pdf, GradNCP significantly outperforms MAML under the given memory constraint, e.g., 66.03$\to$73.45\%. Based on this, we believe exploring GradNCP for general meta-learning setups also has great potential.
>
> ---
>
> **[W1] GradNCP for NeRFs.**
>
> While NeRF is an interesting application of NFs, we primarily focused on achieving higher reconstruction quality of *higher resolution* signals (including image, and video) due to various applications. For instance, a recently emerging paradigm in exploiting NFs, first converts signals into NFs by reconstructing them and casting them for various downstream tasks (e.g., generation or compression [1,2]). In particular, these approaches have shown great promise, e.g., [2] outperforms image, video, and audio codecs in compression, [3] achieves state-of-the-art results in 3D shape generation each of 3D shape data and do generative modeling in this NF weight space (i.e., not NeRF), to name a few. Hence, prior works highlight the importance of the *fast* reconstruction of NFs [1,2,4], where scaling to high-resolution signals was also a challenge.
>
> Nonetheless, we also agree that NeRF is also an important application of NFs. To this end, we apply GradNCP for the NeRF experiment on ShapeNet Car by following the experimental setup in [4]. Here, we train NeRFs with a given single view of the 3D scene and evaluate the performance on other unseen views. As shown in Table 2 in the attached pdf, GradNCP significantly outperforms other baselines including, Learnit (e.g., 22.80$\to$24.06 dB), and even achieves better performance than TransINR which utilizes an additional (large) Transformer. We believe extending GradNCP for scaling large-scale NeRF training (e.g., Google street view) will be an interesting future direction to explore, where we believe the efficiency of GradNCP can contribute in this field.
>
> ---
>
> **[W2-1] Contributions of bootstrapped correction.**
>
> The contributions of bootstrapped correction are mainly two-fold. First, information loss correction by minimizing the distance between the meta-learner and the bootstrapped target adapted with the full context set. This is a unique contribution of GradNCP where we believe it can also be applied to prior methods that use the target to regularize the meta-learner [5] (e,g., use full context for training the target). The second is to further reduce the myopia of the inner optimization, i.e., short-horizon bias, which is known to be the effect of [5]. In particular, achieving better performance in long horizon optimization is crucial, where the joint usage of (i) many inner steps by using reduced memory with context pruning and (ii) bootstrap correction, highly improve the long horizon performance (see Figure 5 (b) in our manuscript).
>
> ---
>
> **[W2-2] Discussion of the gradient norm-based selection.**
>
> Thank you for pointing this out. The gradient norm w.r.t the full parameter is a commonly used selection scheme for data pruning [6], but is very inefficient in our situation that needs to select samples at *every training iteration* (i.e., online). Specifically, it requires the gradient of every individual sample, which is too computationally expensive to compute repeatedly. On the other hand, GradNCP uses the gradient norm w.r.t the last layer which can be efficiently calculated with a single forward pass, thereby enabling to use for online selection. Furthermore, in Figure 4 and Appendix D.1 in our manuscript, we show that our last layer gradient norm and the full gradient norm have a very high correlation, indicating GradNCP is a quite reasonable approximation for sample importance.
>
> ---
>
> **[Q1] Pre-training and finetuning for NFs.**
>
> Unlike conventional meta-learning for classification, standard batch training for NFs is hard to capture the knowledge of the given task distribution. Note that for meta-learning NFs, each data itself (e.g., image) becomes a meta-learning task where each task consists of the same input coordinates with different output signal values. Therefore, standard batch training in NFs performs *regression* on the same input with multiple output targets, which does not lead to a good initialization (e.g., for image, it is a similar effect of training NFs on an averaged pixel value of all CIFAR-10 data). Due to the absence of effective pre-training schemes for NFs, prior research highlighted the importance of meta-learning to learn the initialization of NFs [4].
>
> ---
>
> **Reference**\
> [1] From Data to Functa: Your Data Point is a Function and You Can Treat it Like One, ICML 2022\
> [2] Modality-Agnostic Variational Compression of Implicit Neural Representations, ICML 2023\
> [3] HyperDiffusion: Generating Implicit Neural Fields with Weight-Space Diffusion, ICCV 2023\
> [4] Learned Initializations for Optimizing Coordinate-Based Neural Representations, CVPR 2021\
> [5] Bootstrapped Meta-Learning, ICLR 2022\
> [6] Deep Learning on a Data Diet: Finding Important Examples Early in Training, NeurIPS 2021

---

> > ### Comment · Reviewer_hvG5 · 2023-08-14
> > **Response to Author Rebuttal**
> >
> > I thank the authors for clarifying their technical contributions and the motivation for their work. My concerns have been thoroughly addressed and I will raise my score to accept.

---

> > > ### Author Response · Authors · 2023-08-14
> > > **Thank you for the response**
> > >
> > > Dear reviewer hvG5,
> > >
> > > Thank you for letting us know! We are more than happy to hear that our rebuttal addressed your questions well.
> > >
> > > If you have any further questions or suggestions, please do not hesitate to let us know.
> > >
> > > Thank you very much,\
> > > Authors

---

### Author Rebuttal · Authors · 2023-08-08

Dear reviewers and AC,

We sincerely appreciate your valuable time and effort spent reviewing our manuscript.

As reviewers highlighted, we believe our paper tackles an interesting and important problem (EMYi, 3CuA), and provide an efficient yet effective (all reviewers) framework for meta-learning NFs, validated with extensive evaluations (all reviewers) followed by a clear presentation (all reviewers).

We appreciate your constructive comments on our manuscript. In the attached pdf, we have run the following additional experiments to clarify the reviewer's comments:
- GradNCP for general meta-learning setup, i.e., many-shot classification (Table 1)
- NeRF experiment on ShapeNet Cars dataset (Table 2)
- Ablation study of $L$: number of gradient steps for generating bootstrapped target (Table 3)
- Comparison of loss-based and gradient norm-based top-K selection (Table 4)

We strongly believe that GradNCP can be a useful addition to the NeurIPS community, in particular, due to the enhanced manuscript by reviewers’ comments helping us better deliver the effectiveness of our method.

Thank you very much!\
Authors.

---

### Decision · Program_Chairs · 2023-09-21

**Decision:**

Accept (poster)

**Comment:**

This paper uses meta-learning to improve training of neural fields.  The proposed technique incorporates automated curriculum selection based on examination of per-example gradient norms.  After the rebuttal and discussion phase, all reviewers favor accept, citing convincing empirical results and a clearly developed method.  The AC agrees with the reviewer consensus.